# An interaction network of inner centriole proteins organised by POC1A-POC1B heterodimer crosslinks ensures centriolar integrity

Cornelia Sala [1], Martin Würtz [1,2,4], Enrico Salvatore Atorino[2,4], Annett Neuner[1], Patrick Partscht [3], Thomas Hoffmann [2], Sebastian Eustermann [2] & Elmar Schiebel [1] ✉

Centriole integrity, vital for cilia formation and chromosome segregation, is crucial for human health. The inner scaffold within the centriole lumen composed of the proteins POC1B, POC5 and FAM161A is key to this integrity. Here, we provide an understanding of the function of inner scaffold proteins. We demonstrate the importance of an interaction network organised by POC1A-POC1B heterodimers within the centriole lumen, where the WD40 domain of POC1B localises close to the centriole wall, while the POC5-interacting WD40 of POC1A resides in the centriole lumen. The POC1A-POC5 interaction and POC5 tetramerization are essential for inner scaffold formation and centriole stability. The microtubule binding proteins FAM161A and MDM1 by binding to POC1A-POC1B, likely positioning the POC5 tetramer near the centriole wall. Disruption of *POC1A* or *POC1B* leads to centriole microtubule defects and deletion of both genes causes centriole disintegration. These findings provide insights into organisation and function of the inner scaffold.

Centrioles and basal bodies are ancient, microtubule-based structures present in the protists *Chlamydomonas*, *Tetrahymena* and *Paramecium*, invertebrates, flies and vertebrates[1]. Centrioles play a crucial role in maintaining the integrity and microtubule (MT) organising functions of centrosomes, thereby serving as vital components for cellular organisation, MT-dependent transport processes, and the assembly of the mitotic spindle[2]. Centrioles duplicate once per cell cycle starting in G1/S phase. A new centriole, known as the procentriole, begins to assemble at the proximal end of each of the two existing mother centrioles. The procentriole elongates during G2 and matures by the end of mitosis into a centrosome (centriole with pericentriolar proteins), at which point it disengages from the mother centriole[2]. Upon serum starvation, the older of the two mother centrioles is targeted to the cytoplasmic membrane, where it becomes a membrane-anchored basal body. This basal body nucleates the nine MT doublets of the primary cilium, which plays a crucial role in signal transduction[3].

Due to their shared structure and function, centrioles and basal bodies across various species contain homologous proteins. Through mass spectrometry analysis of basal bodies from *Chlamydomonas* and *Tetrahymena*, the basal body-associated protein POC1 (proteome of centriole 1) was identified[4,5]. Subsequently, POC1 was identified in the hydrozoan *Clytia hemisphaerica*, jellyfish, *Drosophila*, *Xenopus*, Zebrafish, mouse and humans as a component of basal bodies and centrioles[6–9]. While *Chlamydomonas*, *Tetrahymena*, jellyfish and *Drosophila* encode only one *POC1* gene, vertebrates have two *POC1* paralogues, named *POC1A* and *POC1B*[10]. All POC1 proteins have a common structural organisation. The N-terminal region carries a seven blade

[1]Zentrum für Molekulare Biologie der Universität Heidelberg (ZMBH), Deutsches Krebsforschungszentrum (DKFZ)-ZMBH Allianz, Universität Heidelberg, Heidelberg, Germany. [2]European Molecular Biology Laboratory (EMBL) Heidelberg, Heidelberg, Germany. [3]German Cancer Research Center (DKFZ), Heidelberg, Germany. [4]These authors contributed equally: Martin Würtz, Enrico Salvatore Atorino. ✉e-mail: e.schiebel@zmbh.uni-heidelberg.de

WD40 motif followed by a coiled-coil helix connected to the WD40 by a flexible linker. Interestingly, in *Tetrahymena poc1Δ*, cells show a growth defect that is enhanced at elevated temperature[7]. Recent findings suggest that *Tetrahymena* POC1 plays a role in promoting the integrity of centriole triplet MTs as a junction protein that links the B to A and C to B MTs[11]. Although the lack of this stabilisation does not directly impact centriole assembly, it renders the MT triplets more susceptible to mechanical stress, such as basal body bending in response to cilia beating[9,11–13].

Human POC1A and POC1B are very similar in structure, with the exception that the flexible linker connecting the WD40 with the C-terminal coiled-coil is longer in POC1B (Fig. 1a). The human POC1A and POC1B proteins function redundantly in centriole integrity such that only double depletion by siRNA leads to a severe phenotype in mitosis[10]. This functional overlap raises questions about whether POC1A and POC1B share identical or specific binding partners, and which region of the POC1 protein— either the WD40 propeller or the C-terminal coiled-coil—is involved in recognising target proteins. The observation that mutations in human POC1A and POC1B, many of which are located within the WD40 domain, lead to specific diseases, SOFT syndrome (short stature, onychodysplasia, facial dysmorphism, and hypotrichosis) in the case of POC1A[14], and autosomal-recessive cone-rod dystrophy in the case of POC1B[15], suggests that the two human POC1 proteins likely have distinct functions, possibly by binding to a defined set of proteins.

While our understanding of the structure and composition of centrioles and basal bodies is continuously expanding, our knowledge regarding functional domains and the protein inter-action network within centrioles, which contributes to centriole stabilisation, remains relatively limited. What is known so far, is that at the centriole proximal end, the A-C linker connects the A- with the C-MTs of adjacent MT triplets probably providing stability[16]. The protein CEP44 binds to the inside of centrioles to the MT cylinder and interacts with POC1B[17]. Further distally, the inner scaffold extends across the distal portion of the centriole lumen, located beneath the MT triplets of the centrioles. Proteins considered to be inner scaffold components are POC5 (proteome of centriole 5)[18], Centrin that binds to POC5[18], FAM161A (Family with Sequence Similarity 161 Member A)[19], a MT binding protein, POC1B[1] and CCDC15 (coiled-coil domain-containing protein 15)[20]. The human centriole protein WDR90 and its *Chlamydomonas* orthologue POC16 serve as components of the MT wall, playing crucial roles in maintaining the structural integrity of the inner scaffold[21]. WDR90/POC16 localisation to the triplet MT inner junction has been proposed, akin to the suggested localisation of POC1 in *Tetrahymena*[11,21].

Although POC1 proteins play important roles across various species, their specific functions within the lumen of centrioles and how they interact with other inner scaffold proteins to ensure centriole integrity remains poorly understood. Here, we have analysed the role of human POC1A and POC1B in organisation of the inner centriole proteins POC5 and FAM161A[1] and to MDM1[22]. Depending on the particular protein involved, the interaction was facilitated by either the WD40 propeller, the C-terminal coiled-coil, or both regions concurrently. Additionally, we demonstrate that POC1A and POC1B proteins form a heterodimer through their C-terminal coiled-coil domains. Our data suggest an interaction network involving POC1 heterodimers within the centriole lumen, where the WD40 of POC1B localises close to the centriole MT wall, while the POC5-interacting WD40 of POC1A resides towards the centriole lumen. The interaction with the MT binding proteins FAM161A and MDM1 likely tethers the inner scaffold to the wall of centrioles. This interaction network safeguards centriole integrity and compartmentation of centriolar proteins.

## Results

### Domain organisation and localisation of human POC1A and POC1B

Human cells encode *POC1A* and *POC1B*, two closely related paralogues characterised by an N-terminal WD40 domain and a C-terminal coiled-coil connected by a linker region (Fig. 1a). Alpha-Fold2 structure predictions of POC1A and POC1B reveal similar domain organisation for both proteins, with the connecting linker region predominantly disordered, except for a short segment of POC1B, which we have named Intra (Fig. 1a, b). While comparison of the seven WD40 blades and the coiled-coil between POC1A and POC1B demonstrates high similarity, it also highlights, paralogue specific features in all seven blades, particularly noticeable in blades 1, 2 and 4, as well as in the C-terminal region with the coiled-coil (Fig. 1c). Notably, reported mutations in POC1A and POC1B associated with disease primarily occur within the blades of the WD40 domain (Supplementary Fig. 1a), underscoring the crucial role of this domain in the functionality of both proteins. In contrast, the linker region connecting the WD40 domain with the C-terminal coiled-coil in the two POC1 proteins is less conserved, particularly in the Intra segment. This segment, found exclusively in POC1B, shows a propensity to form a beta-strand (Fig. 1b, c). Conservation analysis across species, including invertebrates, reveals a high degree of conservation in both the WD40 domain and the C-terminal coiled-coil regions (Supplementary Fig. 1b, c). To gain a deeper insight into the localisation of POC1A and POC1B and the domain requirements of both proteins, we conducted an analysis of their localisation within the centriole luminal region by Ultrastructure-Expansion Microscopy (U-ExM). Antibodies against the middle/C-terminal (M/C) region of POC1A and POC1B were used for the localisation studies (Supplementary Fig. 2a). The specificity of these antibodies was confirmed as they failed to detect the corresponding protein via indirect immunofluorescence (IF) and immunoblotting in RPE1 *POC1A*−/− and *POC1B*−/− cells (Supplementary Fig. 2b–f). These cells were engineered using a double CRISPR/Cas9 cut strategy targeting the WD40 coding region of the *POC1* genes (Supplementary Fig. 2g–j). U-ExM revealed that in RPE1 cells, the recruitment of POC1A and POC1B to newly formed procentrioles occurs when procentrioles are still relatively short, measuring between ~120 and 140 nm (Fig. 1d). Furthermore, POC1A spanned ~65% of the mother centriole's central region, showing significant overlap in localisation with POC1B. However, POC1B extended slightly further into the proximal end region of mother centrioles compared to POC1A (Fig. 1d–f). In the top views, the middle/C-terminal region (M/C; Supplementary Fig. 2a) of POC1B was observed to be in closer proximity to the centriole MT wall, in contrast to M/C-POC1A, which located more towards the centriole lumen (Fig. 1g, h). The MT-binding proteins MDM1 and CEP44 were located closer to the centriole wall (based on α-tubulin staining) compared to the M/C regions of POC1A and POC1B. POC5 showed a similar positioning to M/C-POC1A, while CCDC15 and FAM161A were positioned between M/C-POC1B and MDM1/CEP44 (Fig. 1i, j). Figure 1k summarises the distances to the MT wall of the analysed inner centriolar proteins, including POC1A and POC1B.

To investigate the regions in POC1A and POC1B necessary for centriole localisation, we expressed various doxycycline (Dox)-inducible *POC1* constructs in the *POC1A*−/− and *POC1B*−/− cell lines. Additionally, we generated WD40 and coiled-coil fragments to assess their potential centriolar localisation as an indicator of their functionality (Fig. 1l). Both POC1A-HA and POC1B-HA were observed to localise to centrioles marked by γ-tubulin (Fig. 1m). In contrast, HA-tagged WD40 domains of POC1 and POC1B (WD40A and WD40B) failed to localise to centrioles, as did the C-terminal regions including the coiled-coil, with these subregions exhibiting diffuse accumulation in the cytoplasm. Intriguingly, chimeric

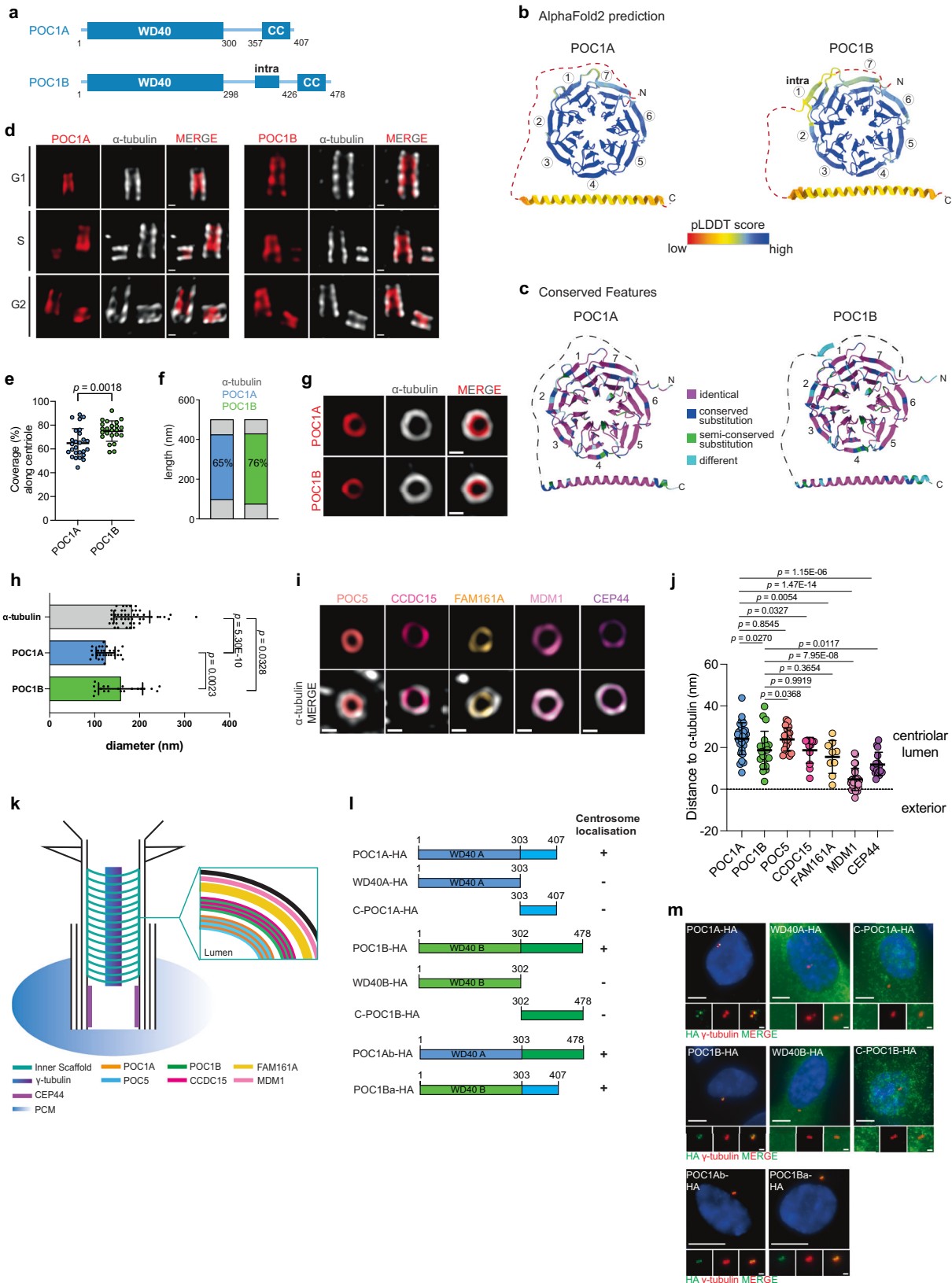

proteins comprising the WD40 domain from one protein and the linker-coiled-coil from the other paralogue (hereafter, POC1Ab or POC1Ba), were able to localise to centrioles (Fig. 1m). Taken together, this suggests that both the WD40 and the C-terminal coiled-coil region are essential for the targeting of POC1A and POC1B to centrioles.

**POC1A and POC1B are required for centriole localisation of inner centriole proteins**

We analysed *POC1A*⁻/⁻ and *POC1B*⁻/⁻ cell lines (Supplementary Fig. 2) in conjunction with conventional IF and U-ExM. Through this approach, we investigated the impact of POC1A and POC1B loss on the localisation of centriolar proteins (Fig. 2 and Supplementary

**Fig. 1 | POC1A and POC1B are luminal centriolar proteins with a similar domain architecture and centriolar localisation. a** Schematic representation of POC1A and POC1B with the WD40 domain at the N-terminus and the coiled-coil (CC) at the C-terminus. **b** Ensembles of the 10 best-ranked AlphaFold2 predictions of POC1A and POC1B. Numbers 1–7 indicate WD40 domain blades. Colouring based on pLDDT score: the linker region (dotted line) has a lower confidence level than the WD40 and CC. **c** Conserved features between POC1A and POC1B in the regions with the higher confidence level. **d** U-ExM images showing the centriolar localisation of POC1A and POC1B in different cell cycle phases. Centrioles were stained with α-tubulin (grey) and POC1A or POC1B (red) antibodies. Scale bars: 100 nm. **e** Quantification of G1 cells from (**d**) measuring the coverage rate of POC1A and POC1B along the centriole. $n = 26$ (POC1A), 24 (POC1B) centrioles. **f** Schematic representation of POC1A (blue) and POC1B (green) along the centriole normalised to centrioles with a length of 500 nm. **g** Top view U-ExM images from centrioles stained with α-tubulin (grey) and POC1A or POC1B (red) antibodies. Scale bar: 100 nm. $n = 28$ (POC1A), 20 (POC1B) centrioles. **h** Diameter analysis of top view centrioles shown in (**g**). **i** U-ExM images from top view centrioles stained against α-tubulin (grey) and the indicated centriolar proteins. Scale bar: 100 nm. **j** Quantification of (**g**, **i**) measuring the distance of the respective proteins to the α-tubulin signal. $n = 28$ (POC1A), 20 (POC1B), 20 (POC5), 12 (CCDC15), 10 (FAM161A), 25 (MDM1), 16 (CEP44) centrioles. **k** Model of the protein organisation within the centriole, comprising the inner scaffold proteins. **l** HA-tagged POC1A and POC1B constructs and their ability to localise to centrosomes. **m** Representative IF images from one experiment of control cells expressing the constructs shown in (**l**). Cells were stained against HA (green) and γ-tubulin (red). Scale bars: 5 μm, magnification scale bars: 1 μm. $N = 3$ biologically independent experiments. **e**, **h**, **j** Data are presented as mean ± SD. All statistics were derived from two-tail unpaired *t*-test. Source data are provided as a Source Data file.

Figs. 3, 4). Because of the centriole luminal localisation of both POC1 proteins, we first focused our analysis on the inner scaffold proteins POC5[18], FAM161A[19], CCDC15[20], Centrin[23,24], and additionally on MDM1[22], that localises along the whole centriole, and WDR90, which is suggested to bridge the inner scaffold with the centriolar wall[21,25].

The absence of POC1A resulted in a significant reduction of POC5 (Fig. 2a, b), FAM161A (Fig. 2a, c), MDM1 (Fig. 2a, d), CCDC15 (Fig. 2a, e) and WDR90 (Supplementary Fig. 3a, b) signal intensities at centrioles to 15%, 50%, 30%, 40% and 70% of control levels, respectively. Contrary to *POC1A*⁻/⁻ cells, the absence of *POC1B* had no significant impact on the signal intensities of POC5, FAM161A and WDR90 at centrioles (Fig. 2a–c and Supplementary Fig. 3a, b). MDM1 and CCDC15 localisation was, however, affected in *POC1B*⁻/⁻ cells (Fig. 2a, d, e). The specific impact of *POC1A* loss on the inner scaffold proteins POC5 and FAM161A was unexpected, given that POC1B, but not POC1A, had been previously linked to inner scaffold function[1]. To confirm these findings, we conducted rescue experiments by Dox-inducible expression of HA-tagged *POC1A* and *POC1B*. Expression of *POC1A* in *POC1A*⁻/⁻ cells successfully restored localisation of POC5 and CCDC15 (Supplementary Fig. 3c–e). Similarly, *POC1B* expression in *POC1B*⁻/⁻ cells resulted in the restoration of MDM1 and CCDC15 centriole localisation (Supplementary Fig. 3f–h).

Subsequently, we employed U-ExM to investigate whether the absence of *POC1A* or *POC1B* influences the distribution of proteins within centrioles. We focused on *POC1A*⁻/⁻ and *POC1B*⁻/⁻ centrioles with near-normal length to exclude that any alterations in localisation were caused by centriole length variations. In both control and *POC1B*⁻/⁻ cells, the inner scaffold proteins POC5 and FAM161A showed a defined central localisation along the MT wall (Fig. 2f–h). However, in *POC1A*⁻/⁻ centrioles, the proper centriole localisation of both proteins was lost (Fig. 2f–h). In control cells, the POC5 interacting Centrin exhibited a strong signal at the tip and a weaker signal in the middle region of centrioles corresponding to the SFI1- and POC5-dependent pools, respectively (Fig. 2f, yellow asterisk and Fig. 2i)[18,23]. Similar to POC5, in *POC1A*⁻/⁻ cells, the weaker Centrin signal in the central region of centrioles was no longer detectable, whereas this Centrin pool remained unaffected in *POC1B*⁻/⁻ cells (Fig. 2f, i). Interestingly, the inner centriole localisation of MDM1 (Fig. 2f, j) and CCDC15 (Fig. 2f, k) was affected by the loss of either POC1A or POC1B. Thus, regarding their dependency on POC1A and POC1B, POC5-Centrin and FAM161A behave differently from MDM1 and CCDC15. Furthermore, the centriole-inside localisation of γ-tubulin and HAUS4[24], was disrupted in *POC1A*⁻/⁻ cells but remained unaffected in *POC1B*⁻/⁻ cells (Fig. 2f, l and Supplementary Fig. 3i, j).

In addition to proteins in the central part of centrioles, we analysed proteins at the proximal end of centrioles (Supplementary Fig. 4) and focused on CEP44, which interacts with POC1B[17], CEP135[26] and the centriole-to-centrosome conversion factor CEP295[27] that associates with the outer PCM part of centrioles and has functions upstream of CEP44 and POC1B[17]. SAS-6 was analysed as an inner centriole specific marker of procentrioles. Compared to control cells, signal intensities showed no significant change in the knockout cell lines, except for CEP135 which is slightly reduced in *POC1B*⁻/⁻ cells (Supplementary Fig. 4a, b). Using U-ExM, however, we observed altered distributions of the proteins along the centriole. In *POC1A*⁻/⁻ cells, the proximal end proteins CEP135 and CEP44, as well as the centriole-to-centrosome conversion factor CEP295, showed a significant extension into the distal domain of centrioles (Supplementary Fig. 4g–j). In contrast, while CEP135 and CEP295 did not show this extension in *POC1B*⁻/⁻ cells (Supplementary Fig. 4g–j), CEP44 exhibited a mild distal extension in those cells (Supplementary Fig. 4g, j). Conversely, SAS-6 localisation remained unaffected in both *POC1A*⁻/⁻ and *POC1B*⁻/⁻ cells (Supplementary Fig. 4k, l). The influence of *POC1A* deletion on CEP135, CEP44, and CEP295 prompted us to investigate whether the loss of other inner scaffold components elicits similar phenotypes. U-ExM of CEP44 in a RPE1 *POC5*⁻/⁻ cell line showed the same phenotype as observed in *POC1A*⁻/⁻ cells: CEP44 extended towards the middle region of the centriole and it lost the defined proximal localisation seen in control cells (Supplementary Fig. 4m, n). Interestingly, the vice-versa experiment, where POC5 localisation was analysed in a published RPE1 *CEP44*⁻/⁻ cell line[17], also revealed an extension of POC5 towards the proximal end of centrioles (Supplementary Fig. 4m, n), suggesting a potential role of centriolar sub-structures in regulating each other's distribution.

In summary, POC1A plays a crucial role in the recruitment and proper compartmentation of inner centriolar proteins POC5, FAM161A, MDM1, CCDC15, and the inner centriolar pool of γ-tubulin and HAUS4. On the other hand, the function of POC1B primarily impacts the localisation of CCDC15 and MDM1.

### The WD40 domain of POC1A binds to POC5

Our analysis revealed mislocalisation of several inner centriolar proteins in *POC1A*⁻/⁻ and *POC1B*⁻/⁻ cells, suggesting potential interactions with POC1 proteins. To explore this possibility, we investigated the binding of POC1A and POC1B with three affected inner centriole proteins. We selected POC5 and FAM161A for this analysis due to their previous identification as inner scaffold components[1]. MDM1 was chosen because its localisation is impacted by the loss of both POC1A and POC1B. To gain structural insights into inner scaffold interactions, we used the AlphaFold2 extension AlphaFold-Multimer[28] and each set of predicted structures was analysed by mapping interacting residues as well as respective confidence scores. An example of this analysis pipeline is shown in Supplementary Fig. 5.

AlphaFold-Multimer consistently predicted interactions of blades 6 and 5 of the WD40 domain of POC1A with residues 472-532 of POC5 (Fig. 3a, bottom and Fig. 3c and Supplementary Figs. 6a, b and 7a, b), while predictions also suggested an additional binding site between

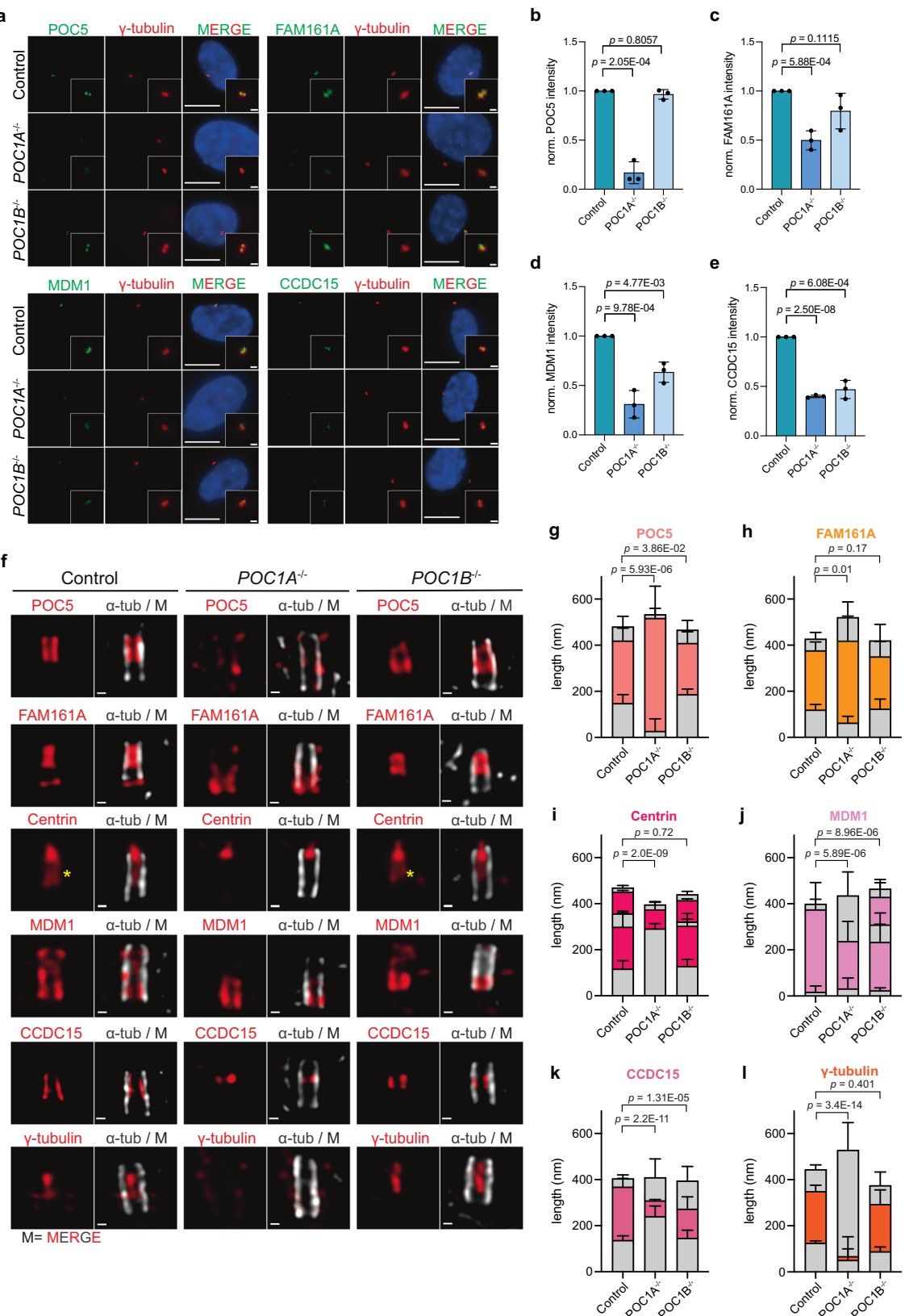

POC5 and blade 1 of the WD40 domain (Fig. 3a, top). AlphaFold-Multimer predicted similar interactions between the WD40 blades 1, 5, and 6 of POC1B and POC5 (Fig. 3b, top; Supplementary Figs. 6a, c and 7c, d). Intriguingly, the predicted Intra beta-strand of the POC1B-specific linker region (residues 361-365), interacts with the extended blade 1 (WD1) of the WD40 propeller (Fig. 3b, bottom;

Supplementary Fig. 7c, d). Intra is predicted to compete with the binding of POC5 to WD1, potentially weakening the interaction between POC5 and POC1B. Additionally, though less robustly predicted, interactions were observed between the C-terminal coiled-coil regions of POC1A and POC1B and regions in POC5 (Supplementary Fig. 7a–d).

**Fig. 2 | Inner scaffold components are affected upon POC1A and POC1B loss. a** IF images of cycling control, *POC1A*[-/-] and *POC1B*[-/-] cells stained against the indicated proteins (green) and the centrosomal marker γ-tubulin (red). Scale bars: 10 μm, magnification scale bars: 1 μm. **b**–**e** Quantification of the signal intensity of the proteins at the centrosome in (**a**). Data are presented as mean ± SD. All statistics were derived from two-tail unpaired *t*-test of *N* = 3 biologically independent experiments, *n* > 100 cells per cell line for each experiment. **f** U-ExM images of intact G1 centrioles (longitudinal view) from control and the respective knockout cell lines stained against α-tubulin (grey) and the indicated proteins (red), M =

merged channels. Scale bars: 100 nm. **g**–**l** Quantification of (**f**). The signal distribution of the respective proteins in each cell line along the centrioles was measured. Data are presented as mean ± SD. All statistics were derived from two-tail unpaired *t*-test. Source data are provided as a Source Data file. **g** *n* = 17 (Control), 10 (*POC1A*[-/-]), 15 (*POC1B*[-/-]) centrioles. **h** *n* = 11 (Control), 11 (*POC1A*[-/-]), 15 (*POC1B*[-/-]) centrioles. **i** *n* = 15 (Control), 14 (*POC1A*[-/-]), 20 (*POC1B*[-/-]) centrioles. **j** *n* = 16 (Control), 10 (*POC1A*[-/-]), 11 (*POC1B*[-/-]) centrioles. **k** *n* = 11 (Control), 10 (*POC1A*[-/-]), 10 (*POC1B*[-/-]) centrioles. **l** *n* = 10 (Control), 11 (*POC1A*[-/-]), 10 (*POC1B*[-/-]) centrioles.

We tested these predictions by co-expressing FLAG-tagged *POC1A* or *POC1B* in HEK293 cells together with HA-tagged *POC5* constructs (Fig. 3c, d) followed by FLAG immunoprecipitation (FLAG IP) experiments. In addition, we examined the WD40 domain and the C-terminal coiled-coil of POC1A/B for interaction with POC5. Consistent with the competition of Intra with POC1B binding to POC5 (Fig. 3b), the IP efficiency between POC1B and POC5 was less compared to POC1A and POC5 (Fig. 3e, compare lanes 5 with 8 and Fig. 3f).

Next, we analysed the binding ability of WD40, the coiled-coil, and POC1Ab and POC1Ba chimeras (in which the C-termini of the POC1 proteins were swapped, see Fig. 1l) to POC5 to gain a deeper understanding of their interactions. Testing the WD40 domains revealed that POC1A[WD40] effectively bound to POC5, while this interaction was absent with POC1B[WD40] (Fig. 3g, compare lanes 5 and 6, and Fig. 3h). Notably, neither the C-terminal coiled-coil region of POC1A nor POC1B exhibited interaction with POC5 (Supplementary Fig. 7e). Furthermore, POC1Ba, which has the C-terminal coiled-coil and linker from POC1A fused to the WD40 of POC1B and thus lacking the Intra region was more efficient in binding to POC5 than POC1Ab, which contains Intra due to the C-terminal coiled-coil/linker of POC1B (Fig. 3g, lanes 7 and 8, and Fig. 3i). This together indicates that the WD40 domain of POC1B binds less strongly to POC5 than that of POC1A, which, along with the intramolecular interaction in POC1B provided by Intra, explains the preferential binding of POC1A to POC5.

As predicted (Fig. 3a and Supplementary Fig. 7a), POC1A did not interact with the N-terminus of POC5 (POC5[1-471]) but instead with the C-terminal POC5[266-575] fragment (Fig. 3e, compare lanes 6 and 7). To further confirm POC5[472-532] binding to POC1A, we expressed *POC5-HA* and *POC5*[Δ472-532]-*HA* lacking amino acids 472-532 together with FLAG-tagged *POC1A* and *POC1B* in HEK293 cells. In FLAG IP experiments, POC5[Δ472-532]-HA failed to bind to POC1A and POC1B (Fig. 3j, compare lanes 2, 3, 5 and 6) confirming POC5[472-532] as the binding site for POC1A and POC1B. This collective evidence suggests that the WD40 domain of POC1A interacts with a fragment located at the C-terminus of POC5.

## POC1A-POC5 interaction is essential for the inner centriole localisation of γ-tubulin
The WD40 domain of POC1A binds to a C-terminal fragment of POC5 (Fig. 3). To further investigate this interaction, we generated a RPE1 *POC5*[-/-] cell line via CRISPR/Cas9 (Supplementary Fig. 8a–e). Similar to *POC1A*[-/-] cells, *POC5*[-/-] cells showed reduced signal intensities of γ-tubulin, FAM161A and WDR90 at the centrosome and loss of the central pool of Centrin in U-ExM (Supplementary Figs. 8d–i and 3a, b). Localisation of POC1A, POC1B and MDM1 was not affected in the *POC5*[-/-] cell line, however, CCDC15 was no longer uniformly distributed along the distal end of the inner centriole MT wall (Supplementary Fig. 8i).

Exploiting *POC5*[-/-] cells, we analysed the importance of the C-terminal fragment in POC5 for the interaction between POC1A and POC5 by expressing either *POC5-HA* or *POC5*[Δ472-532]-*HA* through the addition of Dox in those cells. The expression of full-length *POC5* with a low concentration of Dox (1 ng/ml) was already sufficient for recruitment of the protein to centrioles (Fig. 4a). Conversely, *POC5*[Δ472-532]-*HA* required significantly higher protein expression level for centriole localisation (1000 ng/ml Dox, Fig. 4a–c), indicating a diminished affinity of POC5[Δ472-532]-HA for centrioles.

Previously, it was demonstrated that the proper localisation of γ-tubulin within the lumen of centrioles relies on POC5[24]. In our study, we investigated whether POC5[Δ472-532] retains its functionality in recruiting γ-tubulin to the centriole lumen. Expression of *POC5-HA* successfully restored the localisation of γ-tubulin to the centriole lumen in *POC5*[-/-] cells. In contrast, even with high expression levels induced by 1000 ng/ml, which effectively targeted *POC5*[Δ472-532]-*HA* to centrioles, there was insufficient recruitment of luminal γ-tubulin (Fig. 4d and Supplementary Fig. 8j). This suggests that the POC1A binding site within POC5 plays a crucial role in both the proper localisation of the protein to centrioles and the recruitment of luminal γ-tubulin.

## The C-terminal coiled-coil and the WD40 of POC1 interact with MDM1 and FAM161A
The localisation dependency of MDM1 and FAM161A on POC1A and POC1B (Fig. 2f) identified them as additional interaction candidates for POC1. AlphaFold-Multimer predicted interactions between POC1A and POC1B with MDM1 mainly mediated by the C-terminal coiled-coil (CC) (Fig. 4e and Supplementary Fig. 9a–f). To validate these predicted interactions, we conducted FLAG-IP experiments using full-length and fragments of POC1 as bait and HA-tagged MDM1 as prey. MDM1-HA co-immunoprecipitated with both the full-length POC1A and its C-terminal coiled-coil region (Fig. 4f, lanes 3 and 4). However, no interaction was observed with the WD40 domain of POC1A (Fig. 4f, lane 5). Similar domain interactions were observed with POC1B (Fig. 4f, lanes 7 and 8).

AlphaFold-Multimer predicted an interaction between POC1A and POC1B with FAM161A, involving the WD40 domain and the C-terminal coiled-coil region of both POC1 proteins (Fig. 4g and Supplementary Fig. 10a–f). Subsequent experiments demonstrated that only the full-length POC1A-FLAG and POC1B-FLAG efficiently co-immunoprecipitated with FAM161A, whereas POC1 subdomains failed to bind (Fig. 4h, i). Therefore, the combined presence of the WD40 domain and the C-terminal coiled-coil region in POC1 proteins is essential for stable binding to FAM161A.

Together, the binding analysis with POC5, MDM1, and FAM161A suggests that POC1 proteins exhibit multiple modes of interaction with target proteins, involving both the WD40 domains and the C-terminal coiled-coil region.

## Homo- and heterodimerization of POC1A and POC1B
In our model, through dynamic protein-protein interactions, POC1A and POC1B facilitate the crosslinking of proteins within the centrioles, forming a robust three-dimensional protein network that is crucial for maintaining stability and compartmentalisation. This may involve the homo- and hetero-dimerization of POC1A and POC1B, resulting in the formation of POC1A/B molecules.

AlphaFold-Multimer predicted an interaction between POC1A-POC1B, POC1A-POC1A and POC1B-POC1B primarily mediated by their C-terminal coiled-coil regions (Fig. 5a and Supplementary Figs. 11a–d and 12a–f). Additionally, in the POC1B-POC1B homodimer, interactions were predicted between the WD40 domains and the Intra region of POC1B (Supplementary Fig. 12c, d). Similar to the POC1B homodimer, the POC1A-POC1B heterodimer was also predicted to show interactions between the WD40 domains and the WD40

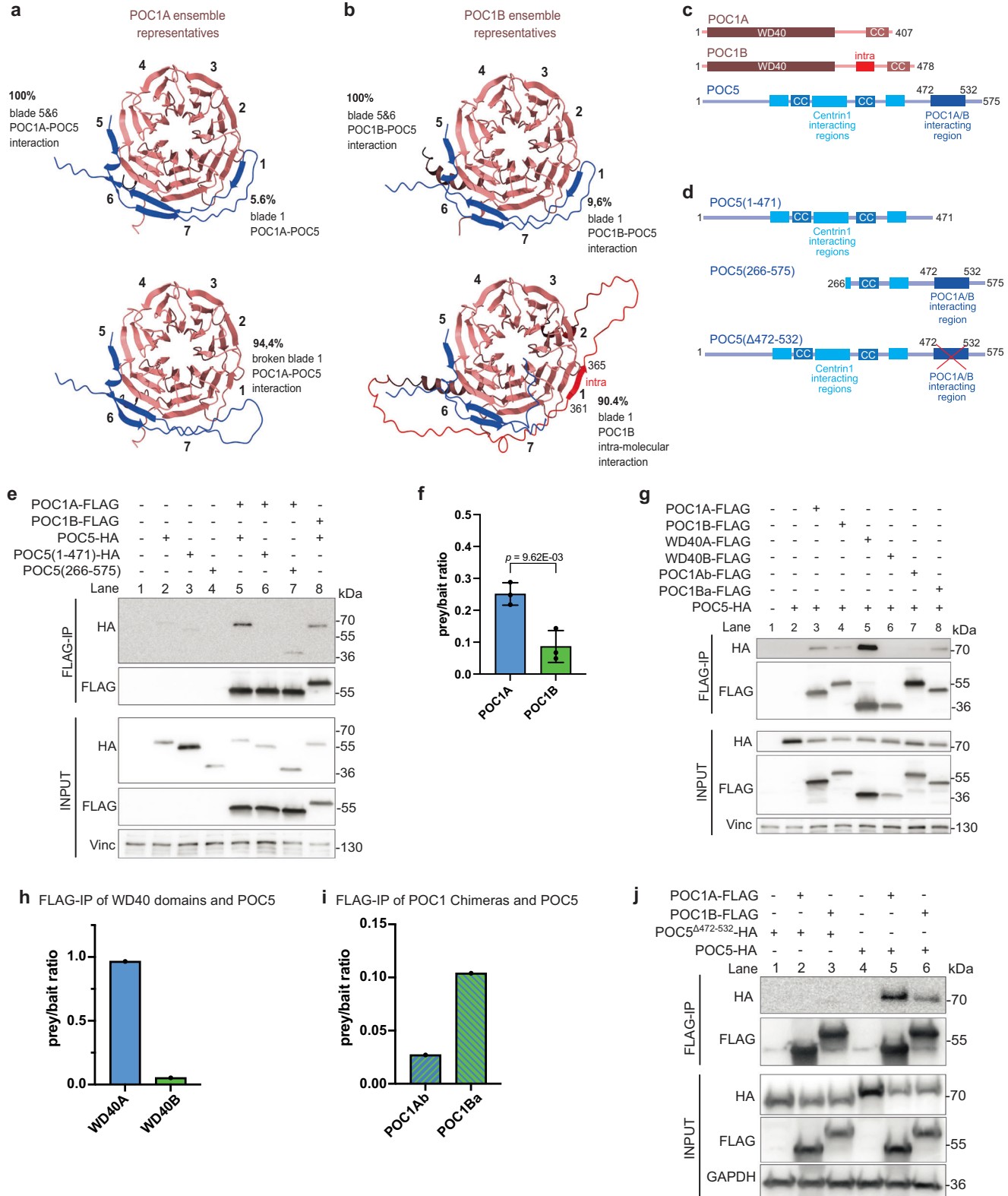

domain of POC1A and the Intra region of POC1B (Supplementary Fig. 12e, f). However, in all cases, the WD40-WD40 interactions were less robust compared to the coiled-coil interactions (Supplementary Fig. 12a, c, e).

Co-immunoprecipitation confirmed interactions between POC1A and POC1B (Fig. 5b, lanes 3 and 13), POC1A and POC1A (lane 10), and POC1B and POC1B (lane 6). These interactions were mediated by the C-terminal coiled-coil (Supplementary Fig. 12g), while the WD40

domains of both POC1 proteins were insufficient to mediate interactions with the other POC1 protein (Fig. 5b, lanes 4 and 11).

We subsequently employed Fluorescence Lifetime Imaging Microscopy (FLIM)-Förster resonance energy transfer (FRET) to validate the proximity of POC1 proteins within centrioles. To accomplish this, we introduced C-terminal tags of mNeonGreen and mScarlet-I to POC1A and POC1B, respectively, in HEK293T cells. The life time of the donor (POC1A-mNeonGreen) in the presence of the acceptor

**Fig. 3 | The WD40 domain mediates the interaction with the inner scaffold protein POC5. a, b** AlphaFold-Multimer predictions of the WD40 domain of the POC1 proteins and POC5 reveal a potential binding site comprising blades 5 and 6 of the β-propeller and to a lesser extend blade 1. The lower panel of (**a, b**) focuses on blade 1, showing loss of the interaction when blade 1 is either broken in the case of POC1A or blocked due to an additional β-sheet (named Intra) from the linker region of POC1B. Percentages indicate the occurrence of the predicted ensembles. Confidence scores are shown in Supplementary Fig. 6. **c** Domain organisation of the POC1 proteins and POC5. **d** Different HA-tagged constructs of POC5 used for immunoprecipitation (IP) experiments. **e** Representative FLAG IP from HEK293T cells expressing either *POC1A-FLAG* or *POC1B-FLAG* together with HA-tagged subdomains of *POC5*. Vinculin (Vinc) was used as input control. **f** Quantification of the FLAG IP shown in (**e**). Due to differences in the expression levels of the FLAG constructs, the signal intensity of the prey band from the IP sample was normalised to the signal intensity of the bait and used as an indication for the binding efficiency between POC5 and the POC1 proteins. Data are presented as mean ± SD. All statistics were derived from two-tail unpaired *t*-test of $N = 3$ biologically independent experiments. **g** Representative FLAG IP from HEK293T cells expressing FLAG-tagged WD40 domain of either POC1A and POC1B or chimeric versions together with HA-tagged full-length POC5. Vinculin is input control. **h, i** Quantification of the prey/bait ratio of the IP samples shown in (**g**). Quantifications show the result from one representative experiment out of $N = 3$ biologically independent experiments. Although the outcomes of the experiments were identical, there was variation between them. **j** FLAG IP from HEK293T cells expressing FLAG-tagged POC1 proteins with HA-tagged full-length POC5 or POC5$^{\Delta472-532}$ to verify the predicted interacting site. GAPDH is used as input control. $N = 2$ biologically independent experiments. **f, h, i** The prey/bait ratio of each experiment and the corresponding immunoblot is provided as a Source Data file.

(POC1B-mScarlet-I) was measured using the analysis tool phasor FLIM (Fig. 5c). In the phasor plot, each pixel intensity in the FLIM image corresponds to a specific localisation: signals with longer lifetimes are positioned on the left, while signals with shorter lifetimes appear on the right. FLIM-FRET analysis in live cells revealed an interaction between POC1A and POC1B, as indicated by a shift (quenching) in the fluorescence lifetime of the donor (POC1A-mNeonGreen) upon the presence of the acceptor POC1B-mScarlet-I (Fig. 5d, e). These data together indicate that the interaction between POC1A and POC1B is facilitated by their coiled-coil domains and that they can in addition to heterodimers, can also form homodimers.

**Induced dimerization of the subdomains partially restores centriolar localisation**

Expression of either the WD40 domain or the C-terminal coiled-coil of POC1A and POC1B alone proves inadequate for establishing centriole localisation in cells (Fig. 1). It appears that combined binding of POC1 subdomains facilitate efficient centriole recruitment. To validate this model, we induced the dimerization of subdomains within cells by expressing Dox-inducible WD40 domains fused to EGFP (POC1A$^{WD40}$-EGFP and POC1B$^{WD40}$-EGFP) and the C-terminal coiled-coil (cc) fused to the GFP-binder protein (GBP) (GBP-POC1A$^{CC}$-mScarlet and GBP-POC1B$^{CC}$-mScarlet, respectively) (Fig. 5f, g). The expression of individual subunits exhibited a dispersed cytoplasmic distribution without co-localisation with the centrosome marker γ-tubulin (Fig. 5h-i-v and Supplementary Fig. 13a, b). Cells transfected with one WD40 domain fused to EGFP together with one GBP-CC construct exhibited an EGFP signal localised at the centrosomes (Fig. 5g, h-vi-ix). Interestingly and consistent with Fig. 1k, the subdomains of the POC1 proteins were interchangeable, as evidenced by the centrosomal localisation observed in cells transfected with either POC1A$^{WD40}$-EGFP/GBP-POC1B$^{CC}$-mScarlet or the POC1B$^{WD40}$-EGFP/GBP-POC1A$^{CC}$-mScarlet pairs (Fig. 5h-vi, viii). In addition, simultaneous expression of WD40 domains from both POC1A and POC1B (POC1B$^{WD40}$-EGFP/GBP-POC1A$^{WD40}$-mScarlet-I) failed to exhibit a distinct centrosomal localisation (Fig. 5h-x-xiv). The outcome of the WD40 dimerization was unchanged when the WD40 domains included the C-terminal linker without the coiled-coil (Supplementary Fig. 13a, b). This indicates that the linkage of the POC1 WD40 and coiled-coil domains is essential for proper localisation at the centrosomes, while WD40 dimerization without the coiled-coil is insufficient for localisation.

**Centrioles of *POC1A* and *POC1B* knockout cells are structurally defective**

Our data unveil a complex interplay between POC1A and POC1B within centrioles, influencing the organisation of luminal proteins. Consequently, the simultaneous absence of both POC1 genes may have a more profound impact on the structural integrity of centrioles compared to the deletion of either gene alone. To elucidate the combined functions of POC1A and POC1B in the lumen of centrioles, we conducted an analysis of centrioles from interphase cells in *POC1A$^{-/-}$*, *POC1B$^{-/-}$* and *POC1A$^{-/-}$/POC1B$^{-/-}$* double knockout cells (*POC1A/B$^{-/-}$*) using thin section electron microscopy. For comparison, control and *POC5$^{-/-}$* cells were included. Longitudinal sections of control centrioles exhibited a structurally intact MT cylinder bordered by distal (DA) and subdistal (SDA) appendages at the distal end of centrioles (Fig. 6a, green and yellow arrow heads). In cross sections, the nine triplet MTs were linked by the A-C linker on the proximal end (Fig. 6a, indigo arrow head) and by the inner scaffold towards the distal end (Fig. 6a, orange arrow head). The average length of centrioles in control cells was determined to be 400 nm (Fig. 6b). In *POC1A$^{-/-}$* and *POC1B$^{-/-}$* cells, most centrioles appeared shorter (Fig. 6b, measuring 200 nm and 175 nm, respectively). In terms of average centriole length, the reduction in *POC1B$^{-/-}$* cells was more pronounced than in *POC1A$^{-/-}$* cells (Fig. 6b). In addition, *POC1A$^{-/-}$* and *POC1B$^{-/-}$* centrioles displayed MT defects in both longitudinal and cross sections (Fig. 6a, magenta arrowheads). Entire MT triplets were absent in *POC1A$^{-/-}$* and *POC1B$^{-/-}$* cells, with this defect most pronounced at the distal end of centrioles and was also observed in *Tetrahymena POC1Δ* mutants[9]. Cross-sections revealed that some DA and SDA were removed from *POC1A$^{-/-}$* and *POC1B$^{-/-}$* centrioles, likely together with triplet MTs (Fig. 6a), a phenotype also observed in *CEP350$^{-/-}$* cells[29]. Notably, we observed that the centriole defects in *POC1A$^{-/-}$* and *POC1B$^{-/-}$* cells closely resembled the aberrations observed in *POC5$^{-/-}$* cells (Fig. 6a). These MT defects in the *POC1A$^{-/-}$* and *POC1B$^{-/-}$* cell lines were confirmed by U-ExM (Fig. 6a). Additionally, IF analysis indicated centriole amplification in *POC1A$^{-/-}$* and *POC1B$^{-/-}$* cells (Supplementary Fig. 13c-e), likely triggered by the de novo centriole assembly pathway[27] and resembles the increased basal body duplication in Tetrahymena *POC1Δ* mutants[30]. Consistent with siRNA double depletion data (Supplementary Fig. 13f-h)[10], *POC1A/B$^{-/-}$* double knockout cells exhibited a severe phenotype, with centriolar structures barely detectable by IF (Fig. 6c–f). The reduction of functional centrioles in *POC1A/B$^{-/-}$* cells was confirmed through IF analysis, using γ-tubulin/PCNT (Fig. 6c, d); PCNT/CEP44 (Fig. 6e, f) as a proximal centriole marker and γ-tubulin/CEP97 (Supplementary Fig. 13i) as a distal centriole marker. Despite the overall reduction in centriole number, cell cycle analysis revealed that the number of centriole signals increased in S and G2 phase *POC1A/B$^{-/-}$* cells compared to G1 cells (Supplementary Fig. 13i), suggesting that some centrioles form via the de novo assembly pathway in S/G2 but then become destabilised in mitosis, similar to observations for *CEP295$^{-/-}$* cells[27]. Consistent with this notion, EM analysis only identified remnants of centrioles in some *POC1A/B$^{-/-}$* cells (Supplementary Fig. 13j).

Despite the absence of fully intact centrioles in *POC1A/B$^{-/-}$* cells, these cells had the ability to form mitotic spindles. Interestingly, at the spindle poles, γ-tubulin and CDK5RAP2 were

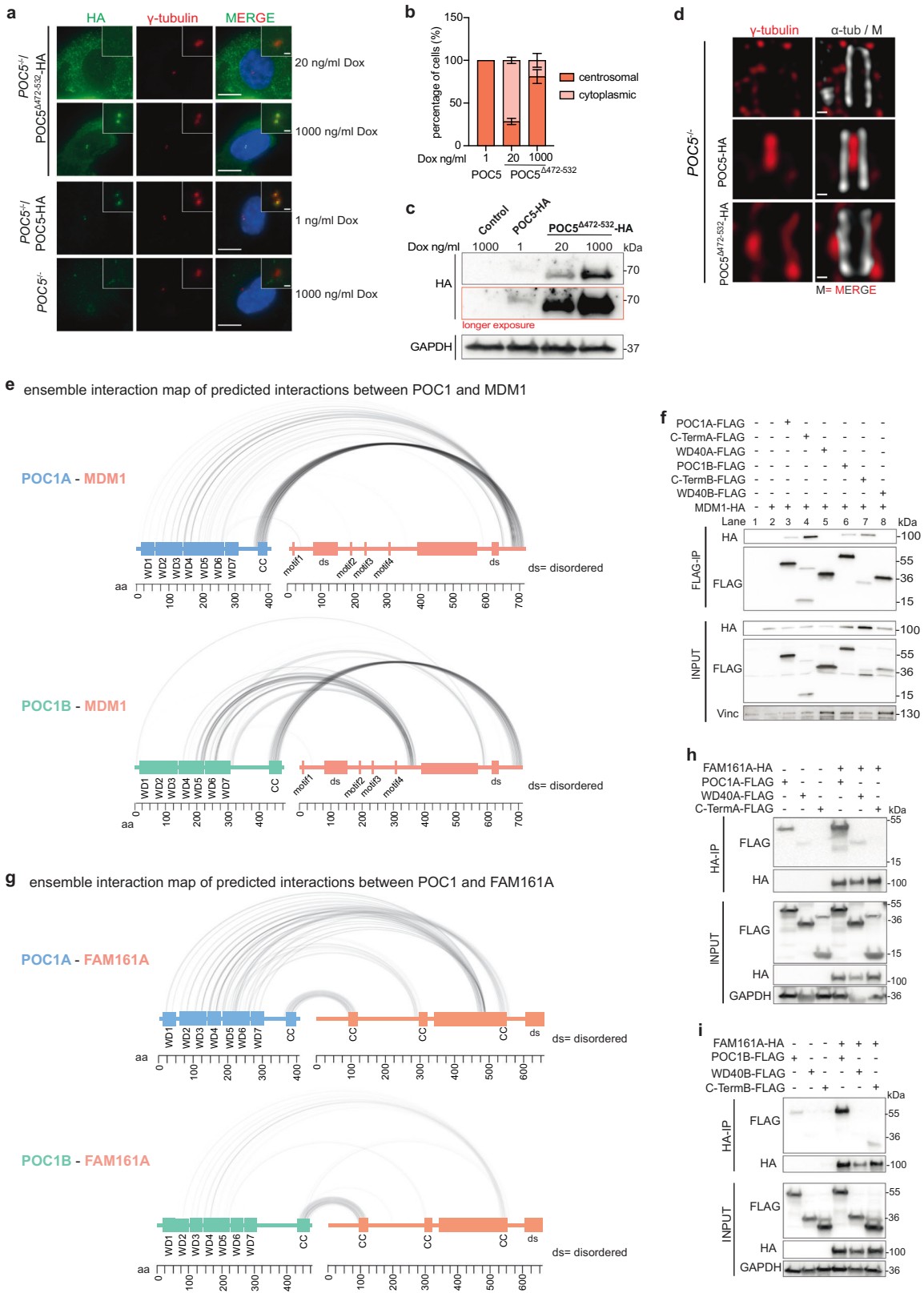

observed (Fig. 6g, h), indicating that an acentrosomal spindle assembly mechanism recruited these proteins[31] or they were derived from centriole remnants. Mitotic spindles in *POC1A/B⁻/⁻* cells often exhibited defects, characterised by the formation of pseudo-bipolar, multipolar, and monopolar spindles. Properly formed bipolar spindles were rarely observed, constituting only

5% of the spindles in *POC1A/B⁻/⁻* cells (Fig. 6g, h). The mitotic defects were less pronounced in the single *POC1* knockouts compared to the double mutant. In conclusion, the enhancement of the centriole phenotypes in the *POC1A/B⁻/⁻* double knockout compared to the single deletions suggests functional redundancy between the two POC1 paralogues.

**Fig. 4 | Different POC1 interaction partners rely on different binding mechanisms. a** RPE1 *POC5⁻/⁻* cells expressing different Dox-inducible versions of HA-tagged *POC5* constructs were checked for centrosomal localisation by IF. Cells were stained against HA-POC5 (green) and γ-tubulin (red). Scale bars: 5 μm, magnification scale bars: 1 μm. **b** Quantification of (**a**). Percentage of interphase cells showing centrosomal and cytoplasmic POC5 localisation. Data are presented as mean ± SD. $N = 2$ biologically independent experiments, $n > 110$ cells per cell line for each experiment. Source data are provided as a Source Data file. **c** Immunoblot of the cell lines from (**a**). The lower HA-immunoblot is a longer exposer of the upper one. GAPDH is used as a loading control. $N = 2$ biologically independent experiments. **d** Representative U-ExM images from intact centrioles of RPE1 *POC5⁻/⁻* cells expressing either full-length *POC5* or *POC5^Δ472-532* and stained against α-tubulin (grey) and γ-tubulin (red), M= merged channels. POC5^Δ472-532 cannot rescue the luminal γ-tubulin localisation. Scale bars: 100 nm. $N = 3$ biologically independent experiments. **e** Ensemble interaction map based on AlphaFold-Multimer predictions of an interaction between POC1A (light blue) or POC1B (green) and MDM1 (salmon) (see Materials and Methods–AlphaFold-Multimer predictions). Interactions predicted to be more robust, appear darker (black) and thicker. The coiled-coil regions of POC1A and POC1B and a C-terminal segment of MDM1 mediate mainly the interactions. aa: amino acids. **f** Representative FLAG IP from HEK293T cells expressing FLAG-tagged full-length or subdomains of either POC1A or POC1B together with HA-tagged MDM. Vinculin is used as input control. $N = 3$ biologically independent experiments. **g** Ensemble interaction map based on AlphaFold-Multimer predictions of an interaction between POC1A (light blue) or POC1B (green) and FAM161A (salmon). The WD40 domain as well as the coiled-coil regions of both POC1 proteins might be involved in the interaction. aa: amino acids. PAE plots and confidence scores for (**e**, **g**) are shown in Supplementary Figs. 9 and 10. **h, i** Representative HA IP from HEK293T cells expressing HA-tagged *FAM161A* and FLAG-tagged subdomains of *POC1A* or *POC1B*. GAPDH was used as input control. $N = 3$ biologically independent experiments.

## A structural model for the POC1A/B-POC5/Centrin network inside centrioles

To understand the role of the POC1A-POC5 interaction, we focused on analysing the structure of the POC5-Centrin2 complex[18]. We employed affinity purification of the POC5-Centrin2 complex reconstituted in insect cells (Supplementary Fig. 14a–c), followed by negative stain EM (Fig. 7a) and mass photometry (Supplementary Fig. 14d). Negative stain EM analysis, coupled with 2D classification and low-resolution 3D reconstruction single-particle averaging, revealed that the POC5-Centrin2 adopted a symmetric elongated structure of about 50 nm that is terminated on both sides by a globular head (Fig. 7a). Mass photometry determined the molecular weight of the POC5-Centrin2 complex to be 418 kDa (Supplementary Fig. 14d), indicating that ca. 6 Centrin2 molecules attach to the POC5 tetramer.

Superimposing the negative stain EM 3D reconstruction of the POC5 tetramer onto the AlphaFold-Multimer prediction of the POC5 tetramer with bound Centrin2, suggests the globular heads in the proximity of the Centrin-binding-regions (Supplementary Fig. 14e, f). This prediction further suggests that the N-terminal coiled-coil from 153-184 of POC5 may play an important role in the formation of a POC5 tetramer. To analyse the role of the amino acid residues 153-184 of POC5 in forming a tetramer, we purified the recombinant POC5^Δ153-184-Centrin2 complex from insect cells (Supplementary Fig. 14g, h). The POC5^Δ153-184-Centrin complex as analysed by negative stain EM exhibited only half the length compared to the WT counterpart and showed only on one side the globular head (Fig. 7b, c). This suggests that POC5^Δ153-184 assembles only into a dimer with bound Centrin2 molecules.

Intriguingly, when *POC5^Δ153-184* was expressed in *POC5⁻/⁻* cells, it exhibited centriole localisation as shown by IF (Supplementary Fig. 14i). This indicates that centriole targeting does not require POC5 tetramerization. However, using U-ExM, POC5^Δ153-184 did not display the proper inner centriole localisation observed with the full-length POC5 (Fig. 7d). Furthermore, *POC5^Δ153-184* was unable to rescue the centriole defects such as mislocalisation of the luminal γ-tubulin pool associated with *POC5* loss, whereas the expression of intact *POC5* successfully restored centriole morphology and luminal γ-tubulin localisation (Fig. 7d).

AlphaFold-Multimer predictions for POC5 from *Paramecium tetraurelia* indicated that it retains similar amino acid properties and forms a tetrameric structure, very similar to that of human POC5 (Supplementary Fig. 14j, k). We modelled the elongated POC5-Centrin structure into the previously published subtomogram averages of centrioles from *Paramecium tetraurelia*[1]. The reconstructed negative stain density from the POC5-Centrin2 complex resembled the density situated in the distal region of the centriole along the longitudinal axis of the MT triplets (Fig. 7e), which might indicate a potential role of the POC5-Centrin2 complex in bridging adjacent MT triplets.

Recent research has proposed a role for *Tetrahymena* POC1 as a junction protein, facilitating linkage between B to A and C to B MTs[11]. To examine the applicability of this model to human POC1A and POC1B, we investigated whether the WD40 domain of POC1 exhibits close proximity to the MT wall of the centrioles. The N-terminal region of POC1A and POC1B, which contains the WD40 domain, was fused with EGFP and the constructs (EGFP-POC1A and EGFP-POC1B) were expressed in RPE1 cells (Supplementary Fig. 14l) to determine the distance of EGFP relative to the MT wall via U-ExM. In control cells, EGFP-POC1B localised close to the MT wall, while EGFP-POC1A localised more towards the lumen of centrioles (Fig. 7f, g). Interestingly, expression of *EGFP-POC1A* in the *POC1A/B⁻/⁻* cell line, changed the localisation of EGFP-POC1A, leading to a localisation in close proximity to the MT wall (Fig. 7f, g). The close localisation of EGFP-POC1B to the centriole MT wall remained unaffected in *POC1A/B⁻/⁻* cells (Fig. 7f, g). Taken together, in control cells the WD40 domain of POC1A is facing towards the centriolar lumen, while the WD40 domain of POC1B is close to the MT wall, which is consistent with recent findings in *Tetrahymena*[11]. In the absence of POC1B, the localisation of EGFP-POC1A is shifting towards the centriole wall.

## Discussion

Although the functional significance of centriole inner scaffold proteins is well recognised, their interactions, role in stabilising centrioles, and connection with the MT wall remain poorly understood. Based on the data presented here, we propose a model for the structure of the inner centriole scaffold that involves the formation of a POC1A-POC1B heterodimer, anchoring other inner scaffold proteins POC5, FAM161A, and CCDC15[1,20] to the centriole wall. POC1B targets the heterodimer to the inner centriole wall (Fig. 7h, step 1), leaving the WD40 domain of POC1A free to interact with the elongated POC5-Centrin complex (Fig. 7h, step 2). FAM161A and MDM1, by binding to the coiled-coil domains of POC1A/POC1B and their ability to interact with MTs[22,32], likely position the POC5-Centrin complex close to the centriole's MT wall (Fig. 7h, step 3). The POC1A-POC1B-dependent positioning of inner centriole proteins is crucial for preserving the structural integrity of centrioles. This POC1A-POC1B anchoring model is supported by POC1A-POC1B heterodimerization, POC5's localisation dependency on POC1A, the preferred binding of POC5 to POC1A, and the distinct spatial localisation of POC1A, POC1B and POC5 inside the centriole lumen.

POC5's preferred binding to POC1A over POC1B is determined by the Intra region of POC1B, which competes with POC5 binding, and by the WD40 domain of POC1B, which is unable to interact with POC5. In *POC1B⁻/⁻* cells, the localisation of POC1A shifts towards the MT wall. We propose that through its ability to form homodimers, the WD40 domain of the POC1A homodimer that is not attached to the MT wall interacts with POC5, explaining why *POC1B* deletion does not affect

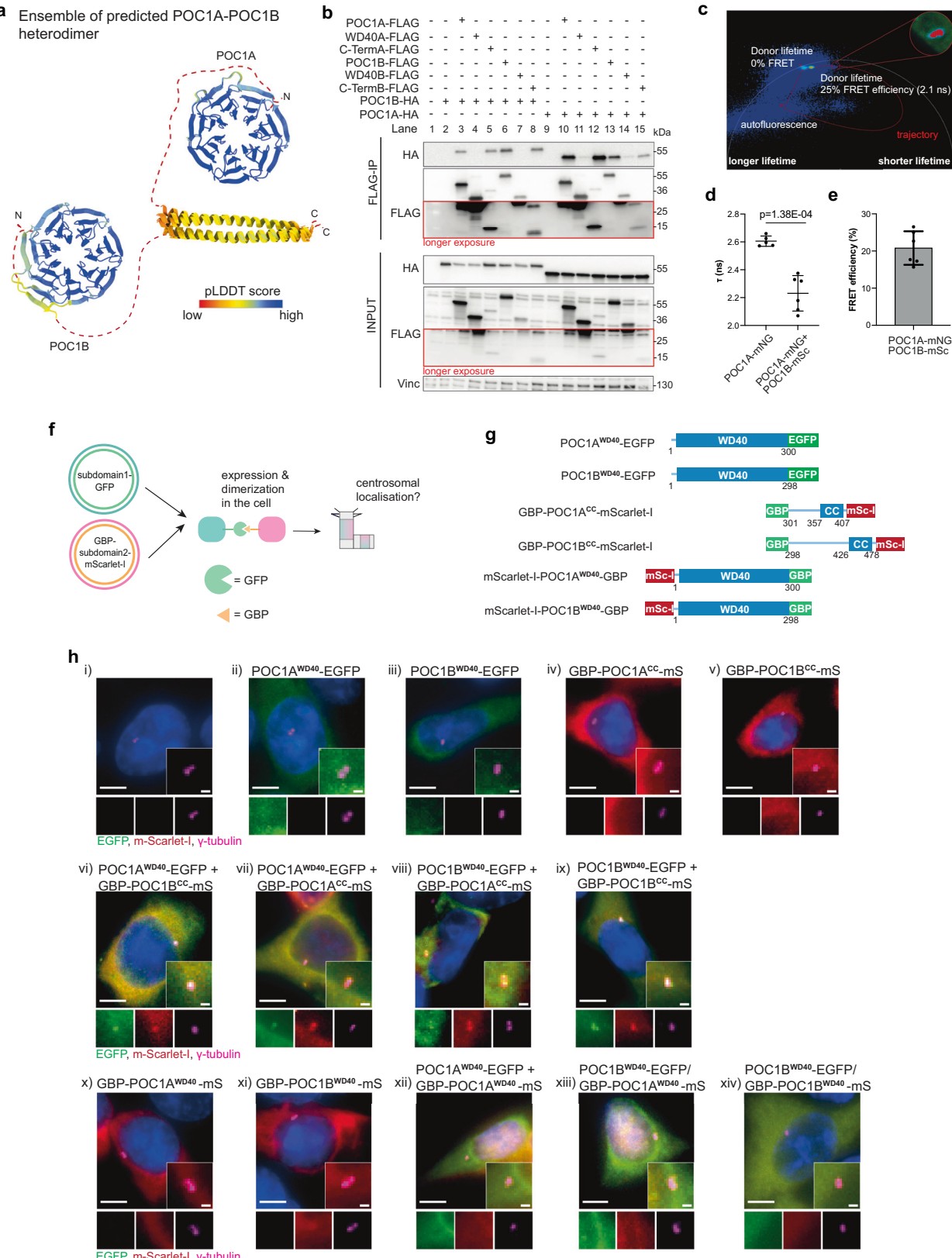

POC5 localisation. POC1B cannot assume this function due to its reduced ability to interact with POC5 and therefore POC5's localisation inside centrioles is affected in *POC1A*[−/−] cells. It is important to note that POC1B in the proximal and distal ends of the centrioles, where it does not overlap with POC1A, may exhibit a different organisational structure compared to the overlapping region of POC1A in the central part of the centriole.

The interaction between POC1A and POC5 is mediated by a region spanning residues 472-532 of POC5, located in its C-terminal flexible part (Fig. 7h, step 2). We show that POC5 forms a tetramer, through its N-terminal region, with multiple Centrins bound to it (Fig. 7h, step 4). POC1A homodimers, as indicated by IP experiments, might connect neighbouring POC5 tetramers on the transversal plane to form the ring-like structure of the inner scaffold (Fig. 7h, step 5).

**Fig. 5 | POC1 proteins form homo- and heterodimers. a** Ensembles of the 10 best ranked AlphaFold-Multimer predictions showing the formation of POC1A and POC1B heterodimers. Colouring based on pLDDT score. The interaction is mediated by the C-terminal coil-coiled region of the POC1 proteins. PAE plots and confidence scores are shown in Supplementary Fig. 11. **b** Representative FLAG IP from HEK293T cells expressing FLAG-tagged full-length or subdomains of either *POC1A* or *POC1B* together with HA-tagged full-length *POC1A* or *POC1B*. Vinculin is input control. N = 3 biologically independent experiments. **c** Representative image of the FLIM-FRET trajectory from the measurement of one cell co-transfected with the Donor:Acceptor pair POC1A-mNeonGreen and POC1B-mScarlet-I. Signals of the FLIM image with a shorter lifetime are located on the right side of the phasor plot, which corresponds to a quenching of the donor signal in the presence of the acceptor. The inset shows the centrosomes and in red are the signals depicted that can be found on the right side of the phasor plot. **d** Quantification of the fluorescence lifetime of the Donor from the representative experiment shown in **c** (N = 2

biologically independent experiments, with n > 5 living cells per condition in each experiment). Data are presented as mean ± SD. Statistics for the representative experiment were derived from two-tail unpaired *t*-test. Source data are provided as a Source Data file. **e** FRET efficiency of the representative experiment shown in **d**. Data are presented as mean ± SD. **f** Experiment scheme for dimerization using the GFP-Binder approach. Subdomains of POC1 proteins were fused either with GFP or with GBP (mScarlet-I functions as a reporter). Upon expression, the subdomains should dimerize via the strong affinity of the GBP for GFP. Centrosomal localisation upon dimerization was checked via IF (see **h**). **g** Constructs used for the GFP-GBP dimerization experiment within the cell. **h** IF of HEK293T cells transfected with the constructs shown in **g**. Centrosomal localisation of the EGFP-tagged (green) and mScarlet-I-tagged (red) proteins were analysed using γ-tubulin (magenta). Additional constructs were analysed in Supplementary Fig. 13a, b. Scale bars: 10 μm, magnification scale bars: 1 μm. N = 3 biologically independent experiments.

Evidence supporting this scaffolding function of POC5 tetramers comes from the observation that tetramerization of POC5 through its N-terminal coiled-coil region is crucial for its functionality. Interestingly, a structure resembling purified POC5-Centrin can be observed by electron tomography in purified centrioles from *Paramecium tetraurelia*[1]. Furthermore, the predicted N-terminal region important for tetramerization of POC5 shows a certain degree of conservation between the species.

The timing of inner scaffold protein recruitment to newly assembled procentrioles varies. Both POC1 proteins are recruited to procentrioles at a length of ~120–140 nm, though it remains possible that one protein is recruited before the other. POC1B interacts with CEP44, which is found early at the procentriole[17], and extends further proximal than POC1A, making it likely that POC1B's recruitment to the centriole is slightly ahead of POC1A's. Unlike POC1, both POC5 and FAM161A have been shown to localise to procentrioles once they reach a length of 160 nm. This may also apply to the central pool of Centrin, which is dependent on POC5 for its recruitment[18,33]. These findings suggest that POC1A/B are recruited earlier than POC5 and FAM161A during procentriole assembly, consistent with the POC1A-POC1B heterodimer being essential for POC5 ansd FAM161A targeting to centrioles. It would be interesting to explore the timing of CCDC15 recruitment relative to other inner scaffold proteins.

Inside the centriole, various substructures, including the cartwheel and the A-C linker, coexist alongside the inner scaffold, each exhibiting distinct lengths. These substructures may act as barriers that restrict the extension of neighbouring structures. In *POC1A*[−/−] cells, which have decreased levels of the MT-binding proteins FAM161A and MDM1 within the centrioles, proximal proteins extend into the central and distal regions of the centriole. The reduction of FAM161A and MDM1 probably exposes free MT-binding sites, allowing them to be occupied by other MT-binding proteins such as CEP44, CEP135, and CEP295, which consequently leads to their extension. A similar phenotype was observed in RPE1 *TUBD1*[−/−] and *TUBE1*[−/−] cells[34]. Centrioles lacking δ- and ε-tubulin contain only A-MTs, fail to recruit POC5, and exhibit an elongated localisation of proximal proteins[34,35]. However, unlike *TUBD1*[−/−] and *TUBE1*[−/−] cells[34], *POC1A*[−/−] cells do not show an extension of the cartwheel protein SAS-6, suggesting a different mechanism for restricting the cartwheel's length. This mechanism might depend on the presence of MT triplets, which can still form in *POC1A*[−/−] cells but are absent in the singlet-MT-containing *TUBD1*[−/−] and *TUBE1*[−/−] cells[35].

The duplication of the *POC1* gene likely occurred early in metazoan evolution, as most vertebrate species possess two *POC1* paralogues, *POC1A* and *POC1B*. As we have shown in this study, the specialisation of POC1A and POC1B enhances specificity, potentially leading to more efficient assembly and maintenance of centrioles. This is particularly important for organisms that rely on numerous cell divisions for development, each of which depends on efficient spindle

formation and functional centrosomes/centrioles. An additional strategy for ensuring proper centriole duplication is the functional redundancy of structural proteins. This helps safeguard the process by preventing mutations in a single gene from completely disrupting centriole duplication. Although *POC1A* and *POC1B* have specialised roles, this study demonstrates that they are also partially redundant in their function, as the simultaneous loss of both genes results in more severe defects than the loss of either gene alone. Additionally, *MDM1* and *FAM161A* may function redundantly in connecting the inner scaffold to the centriole MT wall. Similarly, *POC1* and *WDR16/WDR90* could have overlapping roles in maintaining A-B centriole MT junctions, though further investigation is needed to test this possibility (see below).

Centriole inner scaffold proteins are conserved across a wide range of species, from protists to mammals, which raises the question of whether our model is broadly applicable. While the overall architecture of centrioles is largely conserved among different species, there are significant variations, particularly concerning the cartwheel structure and the binding of inner centriole proteins to the microtubule wall[36]. Additionally, the function of basal bodies and centrioles varies: in most human cells, they primarily act as MT organising centres or basal bodies for signalling through the primary cilium, whereas in protists, they serve as organisers of motile cilia. In this latter role, basal bodies are subjected to significant mechanical stress, making the mechanical modulation of centrioles by inner scaffold proteins crucial for balancing rigidity with plasticity. Given these diverse functional demands, it is not unexpected that inner scaffold proteins exhibit variations across organisms. The single *POC1* gene of *Tetrahymena* is not essential for basal body assembly despite the absence of some B- and C-MTs in *POC1* knockout cells. However, *Tetrahymena* POC1 plays a critical role in stabilising basal bodies when they are subjected to mechanical stress from cilia beating probably because it reinforces the triplet MTs by localising to the A-B and B-C MT inner junctions[11]. It also plays a role in regulating basal body number. Similar defects in centriole MT organisation and centriole numbers were observed in human cells lacking one of the *POC1* genes, despite its paralogue still being expressed and functional. Interestingly, the complete loss of POC1 function in human cells results in centriole collapse, suggesting that the impact of POC1 depletion is more severe in human cells than in *Tetrahymena*. This difference may be attributed to variations in MT-stabilising modifications between organisms or the presence of additional MT-stabilising proteins in *Tetrahymena*.

Based on the doughnut-shaped density resembling a WD40 domain, *Tetrahymena* POC1 is a likely candidate for an inner junction protein[11]. However, centrioles/basal bodies contain with WDR90/POC16 another conserved WD40 domain protein which resides close to the MT wall and loss of WDR90/POC16 function leads to broad inner scaffold defects[21]. Due to its structure and ability to bind to MTs and tubulin, WDR90/POC16 was proposed to be an A-B inner junction

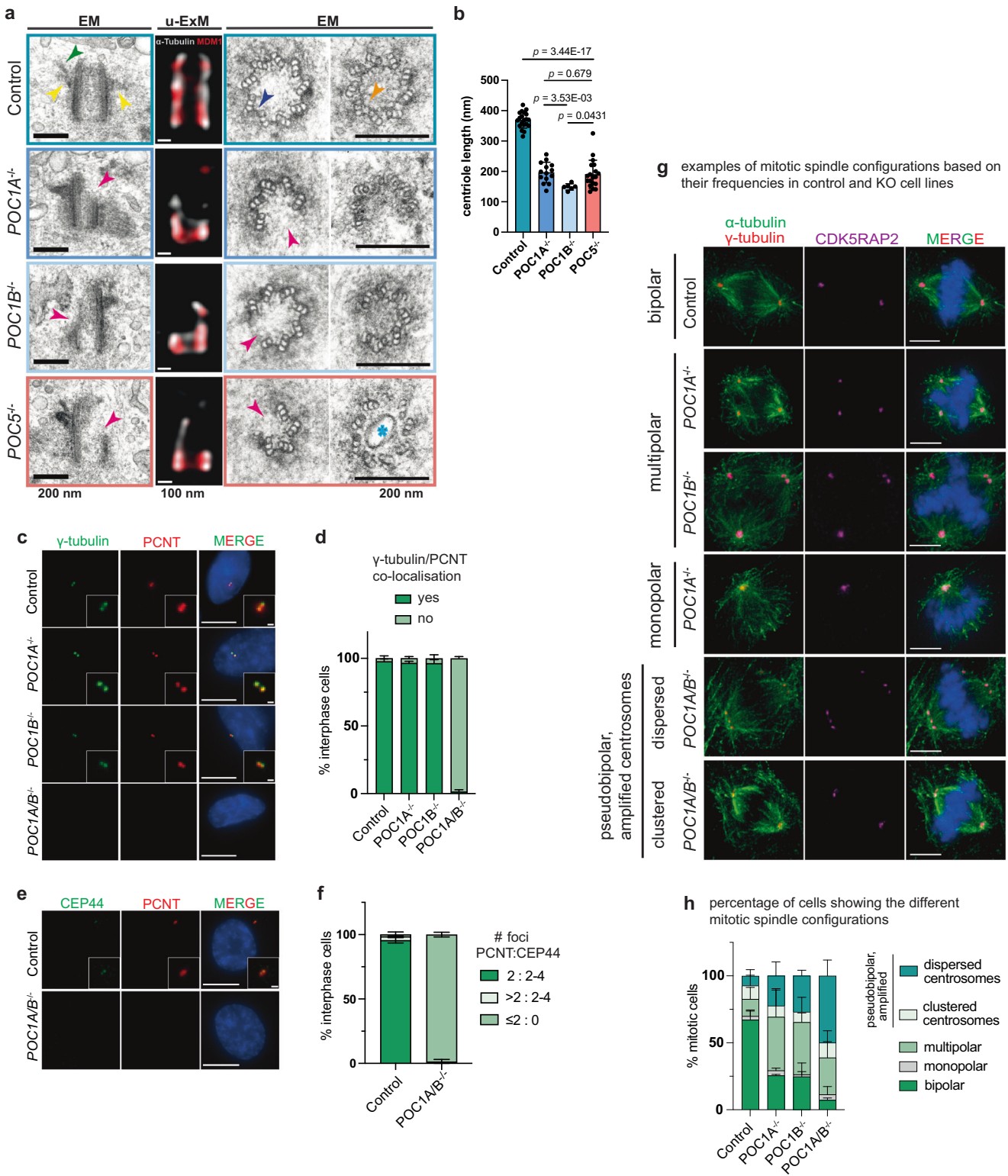

**g** examples of mitotic spindle configurations based on their frequencies in control and KO cell lines

**h** percentage of cells showing the different mitotic spindle configurations

protein that affects centriole integrity and inner scaffold proteins[21,37]. However, the exact role of WDR90/POC16 within the inner scaffold remains unclear, and further experiments are required for clarification. Interestingly, *Tetrahymena POC1Δ* mutant cells still exhibit POC16 localisation at 30 °C with unchanged intensity despite the loss of A-B junction density. This suggests that POC16 may not function as an A-B junction protein, or that while POC1 is not essential for centriole localisation, it is necessary for the localisation of POC16 at the junction[11].

Human POC1A and POC1B dimerize through their C-terminal coiled-coil domains and this ability may extend to POC1 proteins in other organisms as indicated by the conservation of the C-terminal coiled-coil. POC1-POC1 interaction would allow one WD40 domain to bind to centriole MTs while the other recruits inner scaffold proteins. Consistently, in *Tetrahymena* and humans, proper localisation of the inner scaffold protein FAM161A relies on the presence of POC1[11].

*Tetrahymena* Poc5 (TtPoc5) is, contrary to the human orthologue, a transient component of assembling basal bodies, playing a crucial

**Fig. 6 | POC1A and POC1B act together in centriole biogenesis. a** EM images of G1 centrioles from *POC1A⁻/⁻*, *POC1B⁻/⁻* and *POC5⁻/⁻* cells. The knockout cells lines have broken centrioles (as seen in the longitudinal view, magenta arrows) with the most proximal region usually intact (indigo arrow). This phenotype can be also observed by U-ExM. Cross-sections of these centrioles from proximal and distal regions show loss of entire MT triplets (magenta arrows) and in the case of *POC1A⁻/⁻* and *POC5⁻/⁻* deformation or loss of the inner scaffold structure in the distal half of the centrioles. Scale bars: 200 nm (EM) and 100 nm (U-ExM). **b** Quantification of longitudinal centrioles from (**a**). *POC1B⁻/⁻* centrioles show already defects at around 150 nm, whereas *POC1A⁻/⁻* and *POC5⁻/⁻* centrioles have mostly defects at 190-200 nm. Data are presented as mean ± SD. Statistics were derived from two-tail unpaired *t*-test. *n* = 20 (Control), 14 (*POC1A⁻/⁻*), 6 (*POC1B⁻/⁻*), 19 (*POC5⁻/⁻*). **c** IF images of interphase *POC1A/B⁻/⁻* double knockout cells show loss of centrosomal signal. Green: γ-tubulin,

red: PCNT. Scale bars: 5 µm, magnification scale bars: 1 µm. **d** Percentage of the cells from (**c**) showing co-localisation of γ-tubulin and PCNT. Data are presented as mean ± SD. *N* = 3 biologically independent experiments, *n* > 100 cells per cell line for each experiment. **e** Control and *POC1A/B⁻/⁻* cells were stained against CEP44 (green) and PCNT (red) to detect the proximal part of centrioles. Scale bars: 5 µm, and 1 µm for the inset magnifications. **f** Percentage of cells from (**e**) showing CEP44 and PCNT co-localisation. Data are presented as mean ± SD. *N* = 2 biologically independent experiments, *n* > 100 cells per cell line for each experiment. **g** Mitotic spindle configurations observed in control and knockout cell lines. Green: α-tubulin, red: γ-tubulin, magenta: CDK5RAP2. Scale bars: 5 µm. **h** Percentage of cells in the knockout cell lines showing indicated spindle configuration in **g**. Data are presented as mean ± SD. *N* = 2 biologically independent experiments, *n* > 50 cells per cell line for each experiment. Source data are provided as a Source Data file.

role in constructing the distal portion of the basal body structure[38]. The *Tetrahymena* POC5-like gene, *SFR1*, may take over the function for TtPoc5 after its release form centrioles[38]. Despite this difference in recruitment between TtPoc5 and human POC5, human POC5 is also essential for centriole elongation and maturation[18]. Determining whether the POC1-POC5 interaction is conserved in *Tetrahymena* and other organisms is a critical but unanswered question that requires further investigation. If conserved, it would suggest that the fundamental framework of the inner scaffold is shared across species. In conclusion, while some functions of inner scaffold proteins are conserved, comprehensive structural, functional, and interaction studies across various model organisms are necessary for a complete understanding.

## Methods

### Cell culture

Human telomerase-immortalised retinal pigment epithelial cells (hTERT-RPE1) *TP53⁻/⁻*, *CEP44⁻/⁻* [17], *POC1A⁻/⁻*, *POC1B⁻/⁻*, *POC5⁻/⁻* and all stable cell lines derived from these knockouts were cultured in DMEM/F-12 (Gibco) medium supplemented with 10% foetal bovine serum (FBS), 1% L-Glutamine and 1% penicillin−streptomycin. Human embryonic kidney 293 (T) (HEK T293) cells and HEK GP2-293 (Clonetech) cells were cultured in the same medium as RPE1 cells. Cell lines were grown at 37 °C with 5% $CO_2$ and were free of mycoplasma.

### EdU treatment to detect S phase cells

For cell phase analysis, cells were incubated 20 min prior cell fixation with 10 nM EdU from Click-iT™ Plus EdU Alexa Fluor™ Imaging Kit (ThermoFisher Cat. # C10638 and #C10640) following the manufacturer's protocol to detect S phase cells.

### Plasmid transfection and RNAi

Transfection of HEK T293 and HEK GP2-293 was accomplished by using PEI reagent. Plasmid delivery into the RPE1 cell lines was achieved by electroporation using the Neon™ Transfection system 100 µl Kit (Thermofischer Cat. #MPK10096) following the manufacturer's protocol. Transfection of synthetic siRNA oligos were performed using Lipofectamine® RNAiMAX Transfection Reagent from Life Technologies. Transfection reactions were prepared in Opti-MEM™ medium according to manufacturer's protocol. siRNAs used in this study: siPOC1A (5′-CUGGGUACCCAAUGUCAAA-3′; Qiagen, #SI03048871), siPOC1B (5′-GAUUCCGUUGGAUUUGCAA-3′; Qiagen).

### Plasmids and constructs

pRetroX-Tet3G and pVSVG plasmids (Retro-X™ Tet-On® 3 G Inducible Expression System−Clontech) were used to generate RPE1 and HEK T293 cells with Tet-On® 3 G System. Dox-inducible tagged versions of different genes were generated by amplification of the cDNA via PCR and cloned into the pRetroX-TRE3G vector with NEBuilder® HiFi DNA Assembly Master Mix. Constitutive expression constructs were

generated by cloning into the pQCXIP vector with NEBuilder® HiFi DNA Assembly Master Mix. POC1A and POC1B cDNA were obtained as previously described[17] and cDNAs of the other scaffold proteins were obtained from SinoBiological and then cloned into the respective expression plasmids (POC5: # HG25433-UT, FAM161A: # HG22825-UT, MDM1: #HG24563-UT). For the expression of MultiBac constructs in insect cells, *POC5-HA* and *CETN2* (Centrin2) were amplified and cloned into pACEBac1 plasmids. Subsequently, *POC5-HA* was cloned into a pACEBac1 plasmid containing a C-terminal TEV-2xFLAG tag. Gene cassettes were then combined as described previously[39]. To generate *POC5^{Δ153-184}* constructs, the *POC5-TEV-2xFLAG CEN2* construct was amplified using deletion primers, the PCR product was digested with 1 µl Dpn1 at 37 °C for 1 h to remove the template plasmid, and the *POC5^{Δ153-184}-TEV-2xFLAG* plasmid was transformation into DH5alpha. Primers are listed in Supplementary Table 1.

### Stable cell lines and knockout cell lines

Stable cell lines were generated by introducing first the Retro-X™ Tet-On® 3 G Inducible Expression System (Clontech) into the parental RPE1 cell line. The different constructs were then integrated under the TRE3G promotor via Retrovirus transfection (Clonetech).

*POC1A⁻/⁻*, *POC1B⁻/⁻* and *POC5⁻/⁻* cell lines were generated in a *TP53⁻/⁻* background by using a dual sgRNA strategy, in which two different sgRNAs were used to generate a large deletion inside the gene of interest. $1.1 \times 10^6$ cells were electroporated with two plasmids (6 µg per plasmid) containing the sgRNAs cloned into pSpCas9-2 A-GFP (px458, Addgene, #48138[40];. 48 h after electroporation, GFP positive cells were collected via FACS and the pool was initially tested for potential clones via PCR. The pools were then single-sorted into 96-well plates containing complete DMEM-F12 medium and further analysed to obtain single knockout clones of the respective genes. To verify the knockout, DNA from single clones was isolated using the QuickExtraction DNA isolation solution (Epicentre, #QE09050). The isolated DNA was then subjected to PCR to confirm the deletion at the DNA level. Clones showing a deletion were subsequently sequenced and tested by indirect IF and immunoblotting to verify the knockout. Primers are listed in Supplementary Table 1. Target sequences for the knockouts were:

5′-CCGAGATGCAGTTACCTGTG-3′ (POC1A Exon 2),
5′-TCAGGTAGTTTCCCGACGGG-3′ (POC1A Exon 7),
5′-ACTGCATGGGATGGTAACAG-3′ (POC1B Intron 4)
5′-GAAAGGATATCCATAACAGG-3′ (POC1B Exon 10),
5′-GGCTTCCTTGGCGATAACAC-3′ (POC5 Exon 5),
5′-GTAACTGGTAAGGGCATCGG-3′ (POC5 Exon 10).

### Immunofluorescence and signal intensity measurement

Cells were fixed on coverslips with ice-cold methanol at −20 °C for 7 min, washed once with 1xPBS and then blocked in 10% FBS, 0.1% Triton-X100 for 30 min at room temperature (RT). After incubation for 1 h at RT with primary antibody (diluted in 3% bovine serum albumin,

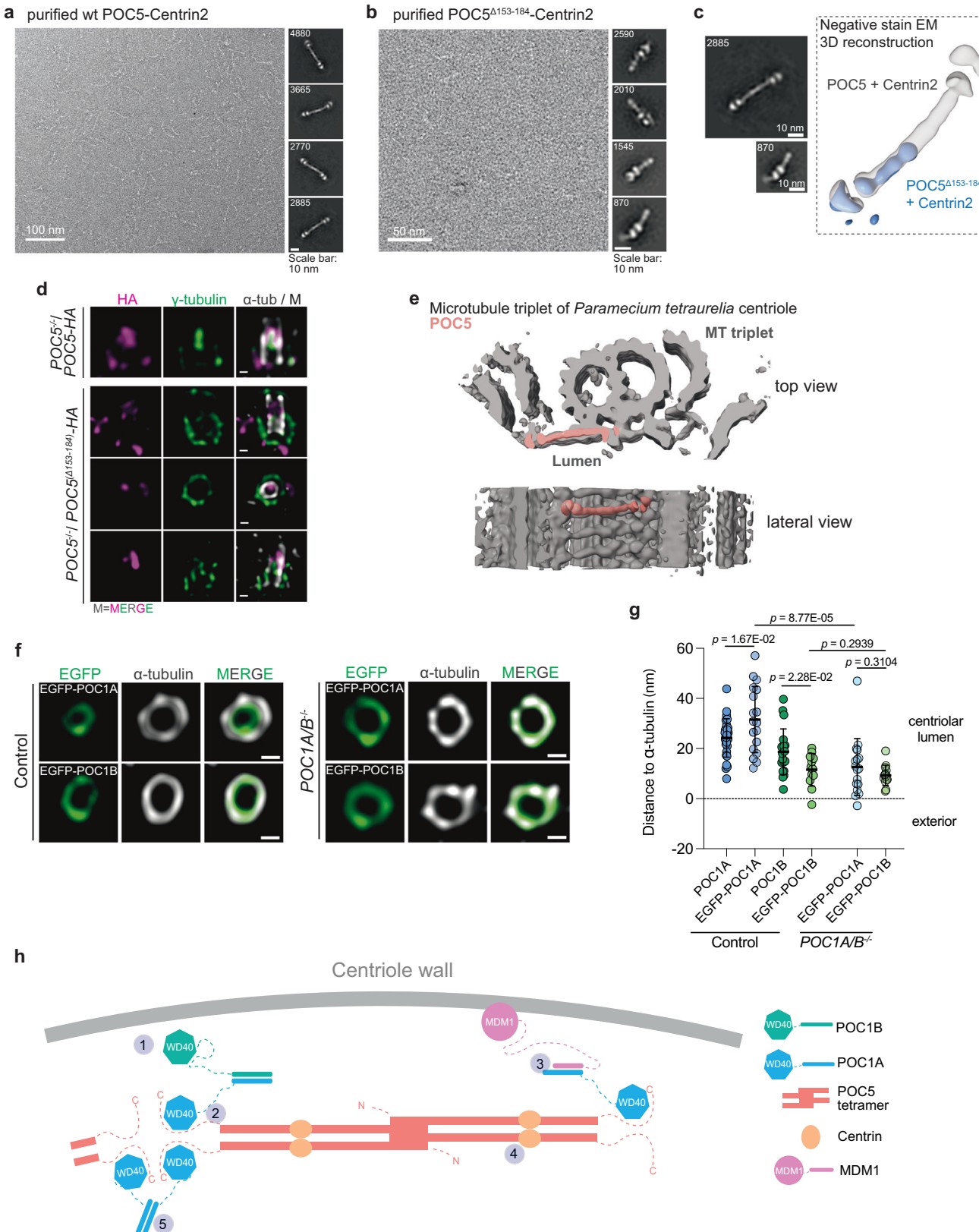

BSA (w/v)), cells were incubated with the secondary antibody (1:500 dilution in 3% BSA) and 4',6-Diamidine-2'-phenylindole dihydrochloride (DAPI) and mounted on glass slides with Mowiol. For antibodies with high background, cells were extracted with CSK-extraction buffer (0.5% Triton X-100, 10 mM PIPES, 300 mM sucrose, 100 mM NaCl, 3 mM MgCl$_2$) prior fixation. The EdU staining was

accomplished after coverslip blocking and before primary antibody incubation, following the manufacture's protocol of the Click-iT™ Plus EdU Alexa Fluor™ Imaging Kit (555 and 647 Kits, ThermoFisher).

All IF images were acquired by the DeltaVision RT system (Applied Precision) with an Olympus IX71 microscope equipped with 60x/1.42 and 100x/1.40 oil objective lenses at RT temperature. Raw images were

**Fig. 7 | An inner luminal protein network ensures the formation of the inner scaffold and the integrity of centrioles.** Representative negative stain Electron Microscopy (EM) micrograph from purified human wild type POC5-Centrin2 (**a**) or POC5$^{\Delta153-184}$-Centrin2 (**b**) complex expressed in insect cells. The numbers in the left upper corner of the 2D class averages indicate particle numbers. Scale bars: 100 nm and 10 nm. $N = 1$ biologically independent experiment. **c** Comparison of wild type POC5-Centrin2 and POC5$^{\Delta153-184}$-Centrin2 complexes. Left: negative stain 2D classes, scale bars and particles numbers are given. Right: negative stain EM 3D reconstructions of wild type POC5-Centrin2 (grey) and POC5$^{\Delta153-184}$-Centrin2 (blue). POC5$^{\Delta153-184}$-Centrin2 particles show a structure shorter than the wild type. **d** U-ExM images of *POC5$^{-/-}$* cells expressing the HA-tagged mutant *POC5$^{\Delta153-184}$* and stained against HA (magenta), γ-tubulin (green), α-tubulin (grey), M = merged channels. Scale bar: 100 nm. $N = 2$ biologically independent experiments. **e** The rod-like POC5-Centrin2 negative stain EM 3D reconstruction (salmon) presented in (**a**, **c**) was fitted into a published[1] subtomogram average of a microtubule triplet from *Paramecium tetraurelia* (emd-4926). **f** U-ExM images of control and *POC1A/B$^{-/-}$* cells expressing POC1A or POC1B tagged at the N-terminus with EGFP and stained against EGFP (green) and α-tubulin (grey). Scale bar: 100 nm. **g** Quantification of centrioles shown in (**f**). The distance between the EGFP signal and α-tubulin was measured from top view centrioles and compared to the distance exhibited when stained with an antibody detecting epitopes at the C-terminus of the POC1 proteins (labelled as POC1A and POC1B, respectively). The data set for centrioles stained with the antibodies detecting the M/C-portions of the POC1 proteins was shown in Fig. 1j and is included in the graph of (**g**) for better comparison. Data are presented as mean ± SD. Statistics were derived from two-tail unpaired $t$-test. $n$ of Control cell lines: 28 (POC1A), 18 (EGFP-POC1A), 20 (POC1B), 12 (EGFP-POC1B) centrioles. $n$ of *POC1A/B$^{-/-}$* cell lines: 17 (EGFP-POC1A), 14 (EGFP-POC1B) centrioles. Source data are provided as a Source Data file. **h** Model of the inner scaffold protein network within the centriole. See Discussion for details.

Z-projected using the softWoRx software provided by the DeltaVision microscope itself (Applied precision). Signal intensity measurements were performed on the maximum projected images in a semi-automated manner using a macro written in Fiji (v 2.0.0/1.52p) that was previously described[41].

### Ultrastructure expansion microscopy (U-ExM)

Cells were grown on 12 mm coverslips and processed as previously described[42]. Briefly, cells were extracted using CSK-extraction buffer and then fixed in a solution with 0.7% formaldehyde and 1% acrylamide diluted in 1x PBS for 3.5 h at 37 °C. Afterwards, gelation was achieved by incubating the coverslips with monomer solution supplemented with 0.5% TEMED and 0.5% APS for 1 h at 37 °C. Subsequently, the coverslips were immersed in denaturation buffer and incubated for 15 min at RT on a shaker to detach the gel from the coverslip. The gel was then put into an Eppendorf tube and boiled in denaturation buffer (200 mM SDS, 200 mM NaCl, 50 mM Tris-BASE and water) for 45 min at 95 °C. After denaturation, the gel was expanded in water for 1 h and then shrank back in 1xPBS. A small piece of the gel was cut out and put in primary antibody solution at 37 °C overnight. The following day, the gel was washed three times with 1xPBS-T before incubating for 3 h at 37 °C in the secondary antibody solution. Lastly, the gel was washed again three times in 1xPBS-T, expanded in water and then placed in a Ibidi μ-Dish 35 mm that was coated with Poly-L-Lysine for imaging.

Image acquisition was performed on an inverted Leica TCS SP8 STED 3× with FALCON FLIM microscope using a HC PL APO 100×/1.40 STED White Oil objectives in RT with a z-interval of 0.15 μm. Raw images were deconvoluted by Huygens' Deconvolution software (SVI Inc.). The z-stack spanning the centrioles were z-projected by ImageJ/ Fiji software.

### Length and diameter measurements of longitudinal centrioles and top-view centrioles

The length and diameter measurements were performed as previously described[1]. Briefly, samples were stained with α-tubulin (as a reference) and the respective protein to have values relative to the reference. Using Fiji, a line scan was drawn along the whole centriole for length measurement and with the plot profile tool and the plugin BAR the fluorescence intensity and the maxima values for each protein were obtained. The length of a signal was considered to be the distance between 50% of the signal intensity of the most proximal and 50% of signal intensity of the most distal peak. The distance of the peaks, which correspond to the length of the respective signals, was corrected by the expansion factor (ranging from 3.75 to 4.2) and then plotted in GraphPad Prism 10. Only centrioles aligned nearly parallel to the x,y plane were considered. Diameter measurements were done as well by using the plot profile tool of Fiji. A line scan was drawn through a top view centriole and the diameter for each protein was obtained by two line scan measurements that were vertical to each other.

Diameters of α-tubulin and the respective proteins for each centriole were normalised to the average α-tubulin diameter from all centrioles and then plotted GraphPad Prism 10. Only centrioles aligned vertically in the z axis were considered for diameter measurements.

### FLIM-FRET microscopy

FLIM-FRET microscopy was performed with live HEK293T cells co-transfected with mNeonGreen-tagged POC1A (donor) and m-Scarlet-I-tagged POC1B (acceptor) on a Leica TCS SP8 STED 3× microscope with FALCON FLIM with white light laser (WLL) and a HC PL APO 100×/1.40 STED White Oil objective in room temperature. Fluorescence lifetime was measured using the FLIM option in the Leica LAS X software v.3.5.7 (Leica Application Suite X). Images were acquired with an imaging repetition of 1000 photons/pixel. Data analysis was done directly in the LAS X software using the phasor plot tool to draw a FRET trajectory and to extract fluorescence lifetime values and FRET efficiencies.

### Electron microscopy

Cells were seeded on coverslips and cultured at 37 °C and 5% $CO_2$. By reaching a confluency of ~80– 90%, cells were washed three times with phosphate buffer (PBS) and then pre-fixed with a mixture of 2.5% glutaraldehyde, 1.6% paraformaldehyde, 2% sucrose in 50 mM caco-dylate buffer for 30 min at RT. After rinsing five times with cacodylate buffer, cells were post-fixed with 2% $OsO_4$ for approximal 45 min on ice, in darkness. Cells were washed then four times with distilled $H_2O$ and incubated overnight at 4 °C in 0.5% aqueous uranyl acetate. On the following day coverslips were rinsed again 4 times with $dH_2O$ and subsequently stepwise dehydrated with ethanol. Cells on coverslips were immediately placed on capsules filled with Spurr-resin (Sigma-Aldrich) and polymerised at 60 °C for ~2 days. In resin embedded cells were sectioned on a Reichert Ultracut S Microtome (Leica Instruments, Vienna, Austria) to a thickness of ~80 nm. Post-staining with 3% aqueous uranyl acetate and lead citrate was performed. Serial-sections were viewed at a Jeol JE-1400 (Jeol Ltd., Tokyo, Japan), operating at 80 kV, equipped with a 4k x 4k digital camera (F416, TVIPS, Gauting, Germany). Micrographs were adjusted in brightness and contrast applying Fiji software.

### Protein purification

For protein expression in insect cells, protocols described earlier[39] were used using the MultiBacTM system. Briefly, the *POC5-CETN2* construct was transformed into DH10MultiBac cells, and v0 baculo-viruses were produced in Sf9 cells using Cellfectin II (Invitrogen), following the manufacturer's instructions. After 72 h, v0 baculoviruses were harvested and used to infect 30 ml of Sf9 cells at a density of $1 \times 10^6$ cells/ml. For protein expression, v1 baculovirus was diluted 1:50 in the expression volume at a cell density of $1–2 \times 10^6$ cells/ml Sf21 cells. Proteins were expressed for 60 h, harvested via centrifugation ($800 \times g$ for 5 min), snap-frozen in liquid nitrogen, and stored at −80 °C until

further use. The expression was performed in SF-900 III medium (Thermo Fisher Scientific) supplemented with 100 units/ml penicillin and 100 µg/ml streptomycin (Thermo Fisher Scientific).

Cells were resuspended in cold lysis buffer (50 mM Tris, pH 7.5, 200 mM NaCl, 1 mM MgCl$_2$ and 1× tablet EDTA-free protease inhibitor tablet (ROCHE), sonicated (3 × 1 min, 0.6 amplitude) and centrifuged at 20.000 × $g$ for 30 min at 4 °C. Afterwards, the lysate was incubated with anti-FLAG M2 Affinity Gel Beads (Sigma-Aldrich) for 1 h at 4 °C while rotating. Beads were then washed 1× with lysis buffer and 2× with wash buffer (50 mM Tris, pH 7.5, 150 mM NaCl, 1 mM MgCl$_2$, 1 mM EGTA, 0.5 mM DTT). For elution, beads were incubated for 30 min at 4 °C in elution buffer (wash buffer, 0.2 mg/ml 3xFLAG peptide) while rotating. Elutions were used either directly for SDS-PAGE or loaded prior onto an SEC column. SEC runs were performed in SEC buffer (50 mM Tris, pH 7.5, 150 mM NaCl, 1 mM MgCl$_2$) using Äkta Pure/ ÄktaGo controlled via Unicorn (v.7.5/7.9). For full-length POC5/ Centrin2 samples the Superdex® 6 (Cytiva) Increase was used, whereas for the POC5$^{\Delta153-184}$/Centrin2 samples the Superdex® 75 column (Cytiva) was used. After SEC, samples were snap frozen in liquid nitrogen and stored at −80 °C until further usage.

## Negative stain EM
For negative stain EM, 5 µl of the sample was applied to glow-discharged copper-palladium 400 mesh grids covered with an ~10 nm thick continuous carbon layer (G2400D, Plano GmbH). After a 30 s incubation at RT, the grids were blotted with Whatman filter paper 50 (CAT N. 1450-070) and washed with three drops of water. The sample on the grids was stained with 3% uranyl acetate in water. Images were acquired using a Talos L120C transmission electron microscope (TEM) equipped with a 4k × 4k Ceta CMOS camera (Thermo Fisher Scientific). Data acquisition was performed using EPU software (v2.9, Thermo Fisher Scientific) at a nominal defocus of ~−2 µm and an object pixel size of 0.2552 nm.

Image processing was conducted in Relion 3.1 for all datasets. The contrast transfer function (CTF) of micrographs was estimated using Gctf. Initially, ~500 particles were manually selected to create an initial 2D class for automated particle picking. Following automated particle picking, 2D classification was performed into 20 − 200 classes, using a T-factor of 2, a translational search range of 20 pixels with a 2-pixel increment, and a mask diameter of 300 − 650 Å. Using the same mask diameters, 3D classifcations were perfomed using T-factor of 4, with a angular sampling interval of 7.5°, a translational search range of 20 pixels and a search step of 2 pixels. Particles were classified into 2 or 3 classes. 3D refinement and post-processing were perfomed with default parameters with solvent masks created from 3D refinment run. For the POC5-Centrin2 wild-type dataset, 521 micrographs were acquired, resulting in 109,299 particles being picked automatically and extracted at full pixel size. Extracted particles were sorted through two subsequent rounds of 2D classification, retaining the best true positive classes. A total of 50,652 particles were used to generate an initial model, which served as a template for 3D classification. After one round of 3D classification, 31,687 particles were selected for 3D refinement and post-processing. For the *POC5*$^{\Delta153-184}$-*CETN2* construct (351 images), 68,238 particles were picked automatically and subjected to three consecutive runs of 2D classification at full spatial resolution. In total, 32,172 particles were selected to generate an initial model, which was used as a template for 3D classification. Following another round of 2D classification, the best classes were selected, resulting in 19,021 particles used for 3D refinement and post-processing, using a 3D reference from the 3D classification.

## Mass photometry
High-precision microscope coverslips (24 × 50 mm) were washed with ddH$_2$O, isopropanol, ddH$_2$O, isopropanol, ddH$_2$O and dried under a pressurised air stream. A silicone gasket with 6 cavities was placed on

top/centre of the coverslip to form measurement holes. A total amount of 19 µl of buffer ((SEC buffer) for both protein samples was applied in each gasket hole and autofocus was performed before every measurement. One microliter of the protein at a concentration of 400 nM was mixed with the SEC buffer. All measurements were performed using a Refeyn TwoMP mass photometer (Refeyn Ltd, Oxford, UK). Videos of 1 min with regular image size were recorded using the Refeyn AcquireMP 2024 R1 software (Refeyn Ltd, Oxford, UK) and data analysis were performed using the Refeyn DiscoverMP 2024 R1 software (Refeyn Ltd, Oxford, UK). Bovin serum albumin (BSA, 66 kDa) and Immunoglobulin G (IgG, 150 kDa and 300 kDa) proteins were used to generate the standard contrast-to-mass calibration curve.

## Antibodies
Primary antibodies used in this study for IF and expansion microscopy were: γ-tubulin (mouse, 1:1000, abcam Ab27074), γ-tubulin (guinea pig, 1:50, homemade), PCNT (rabbit, 1:2000, abcam Ab4448), PCNT (guinea pig, 1:800, homemade), CEP97 (rabbit, 1:300, Bethyl A301-945A), CDK5RAP2 (rabbit, 1:500, Merck 06-1398), Centrin (mouse, 1:1000, Millipore MABC544), Centrin (rabbit, 1:500, Abcam ab101332), α-tubulin (mouse, 1:500, SigmaAldrich DM1A), α-tubulin (rabbit, 1:500, Proteintech 11224-1-AP), α-tubulin (mouse, 1:500, Proteintech 660311-1-Ig), HA tag (rat, 1:1000, Merck 11867423001), GFP (mouse, 1:1000, Roche 11814460001), Mitosin (mouse, 1:100, BD 610768) POC1A (guinea pig, 1:200, homemade[17],), POC1A (rabbit, 1:300, PA5-59217, ThermoFisher), POC1B (guinea pig, 1:500, homemade[17]), POC1B (rabbit, 1:250, PA5-24495, ThermoFisher), POC5 (rabbit, 1:1000, Bethyl A303-341A-T), FAM161A, (rabbit, 1:500, Sigma HPA-032119), WDR90 (rabbit 1:250, NovusBio, NBP2-31888), MDM1 (rabbit, 1:500, PA5-59638, ThermoFisher), CCDC15 (rabbit, 1:1000, ThermoFischer, PA5-59184), HAUS4 (rabbit, 1:500, Proteintech 20104-1-AP), CEP295 (rabbit, 1:500, Abcam Ab122490), CEP135 (rabbit, 1:200, homemade[43]), CEP44 (rabbit, 1:500, homemade[17]).

Primary antibodies used in this study for IB were: FLAG tag (rabbit, 1:1000, Proteintech 20543-1-AP), HA tag (rat, 1:1000, Merck 11867423001), HA tag (rabbit, 1:1000, Proteintech 51064-2-AP), GAPDH (mouse, 1:1000, Proteintech 60004-1-lg), Vinculin (mouse, 1:5000, Proteintech 66305-1-lg), POC1A (guinea pig, 1:200, homemade[17],), POC1A (rabbit, 1:300, PA5-59217, ThermoFisher), POC1B (guinea pig, 1:500, homemade[17],), POC1B (rabbit, 1:250, PA5-24495, ThermoFisher), POC5 (rabbit, 1:1000, Bethyl A303-341A-T).

Secondary antibodies used in this study for IF and expansion microscopy were: Anti-mouse IgG Alexa Fluor 488/555/647 (donkey, 1:500, ThermoFisher), Anti-rabbit IgG Alexa Fluor 488/555/647 (donkey, 1:500, ThermoFisher), Anti-guinea pig IgG Alexa Fluor 488/555/647 (goat, 1:500, ThermoFisher), Anti-rat IgG AlexaFluor 488/647 (donkey, 1:500, ThermoFisher), Anti-rabbit IgG Abberior STAR635P (goat, 1:500, Abberior) and Anti-mouse IgG Abberior STAR635P (goat, 1:500, Abberior). Secondary HRP-conjugated antibodies used in this study for IB were purchased from Proteintech and used in a 1:5000 dilution.

## Protein pull-down and western blot
In order to test POC1 protein interactions, the various *POC1-FLAG* constructs were co-expressed with the potential HA-tagged interaction partners under the control of the Dox-inducible promotor. Briefly, HEK T293 were transfected with PEI, induced with dox, harvested after 24 h and lysed with 150 mM NaCl, 10 mM Tris-Cl, 0.5 mM EDTA, 0.5% NP-40, 1 mM PMSF, 10 U/µl Benzonase, 1 tablet per 10 ml Roche protease inhibitor cocktail complete (EDTA free), pH 7.5 buffer. The lysate was then incubated with the beads for 1 h while rotating at 4 °C. After three washing steps with 300 mM NaCl, 10 mM Tris-Cl, pH 7.5 buffer, the beads were incubated with 30 µl elution buffer containing 150 mM NaCl, 10 mM Tris-Cl, 0.5 mM EDTA and 0.2 mg/ml FLAG peptide in the case of FLAG-IP and then boiled with 4x Laemmli buffer for 5 min at

95 °C. For HA-IP, beads were directly boiled with 30 µl of 4x Laemmli buffer. The samples were analysed by IB, by running SDS-PAGE and transferring to a PVDF membrane. The membranes were blocked in 5% w/v non-fat dry milk in 1xTBS-T (TBS, 0.1% Tween 20) for 1 h at RT and afterwards incubated with indicated primary antibodies at 4 °C. The following day, the membranes were washed with 1xTBS and incubated with secondary antibodies for 1 h at RT and then washed again before imaging. The Clarity and Clarity Max ECL Western Blotting Substrates (Bio-Rad) were used to develop the membrane. Imaging was performed using a Fujifilm LAS-4000 system. Uncropped immunoblots are provided as a Source Data file.

### AlphaFold2 and AlphaFold3 predictions

Predictions were performed with a customised multimer pipeline of AlphaFold2 (AF2, release version 2.3.2)[28,44]. For each candidate sequence, multiple sequence alignments (MSA) were computed separately. The MSAs were combined to be passed to features extraction using the AF2 multimer_v3 parameter set and the UniRef30 database version 2023_02. The maximum template release date was set to the future, default values were used for db_preset (full_dbs) and num_recycle (20). For each multimer candidate, 25 models were predicted in a separate compute job for each model using an individual random seed. All models were passed to the AF2 Amber relaxation.

For the prediction of the POC1A/B WD40 domains with POC5, the latter was cropped to the interacting region of residues 472–532. Ensembles of 125 predictions (num_multimer_predictions_per_model = 25) were computed with AF2 using the procedure described above, but the relaxation pipeline step was omitted.

Possible inter-chain interactions determined were with the FoldX (version 5.0) software[45] for each predicted ensemble and weighted due to plDDT and PAE scores. Additionally, intra-chain interactions were determined between region of POC1A/B residues 309/308 to 407/478 containing intra and 1-478.

The prediction of a POC5 tetramer from *Paramecium tetraurelia* was performed using the AlphaFold Server (powered by AlphaFold3)[46].

### Statistics and reproducibility

Signal intensity measurements of IF images were conducted on maximum projected images using Fiji/ImageJ. Information on the statistical analysis of the experiments is provided in the figure legends. Data are derived from at least three biologically independent unless specified otherwise. Graphs from U-ExM experiments were derived from pooled values of independent experiments. For Western Blot analysis, the band intensity was measured using Fiji/ImageJ. Due to differences in expression levels of the POC1 constructs, the band intensity of the prey protein was normalised to the bait intensity. For statistical significance tests, unpaired two-tailed student $t$-tests and one-way ANOVA were used to test the statistical significance between the sample conditions with a significance level of $p \leq 0.05$. Statistical analysis was done using Prism (GraphPad Prism v10.2.3) and Microsoft Excel (v16.90). No statistical method for data exclusion or predetermination of sample size was perfomed. Experiments were not randomised and investigators were not blinded.

### Reporting summary

Further information on research design is available in the Nature Portfolio Reporting Summary linked to this article.

## Data availability

All data that support the findings of this study are provided with this paper. The data underlying all the quantification presented in this work are available in the Source Data file and the Supplementary Information. All other relevant data supporting the key findings of the study are available from the corresponding author upon request. Source data are provided with this paper.

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

## Acknowledgements

We are thankful to Dr. H. Lorenz of the ZMBH Imaging Facility for his assistance with the FRET analysis. Additionally, we acknowledge Dr. M. Langlotz from the ZMBH FACS Facility for their support in sorting GFP-positive Cas9 transfected cells, Dr. B. Martin from the ZMBH IT department for his help in the analysis of AlphaFold2 predictions and Dr. Karine Lapogue from the European Molecular Biology Laboratory (EMBL) Protein Expression and Purification Core Facility (PEPCF) for performing the Mass Photometry. We acknowledge the access and services provided by EMBL IT Services. We acknowledge the services SDS@hd and bwHPC supported by the Ministry of Science, Research and the Arts Baden-Württemberg, as well as the German Research Foundation (INST 35/1314-1 FUGG and INST 35/1134-1 FUGG). We also acknowledge access to the infrastructure of the Cryo-EM Network at the Heidelberg University (HDcryoNET) and support by Bram Vermeulen and Stefan Pfeffer. We thank Dr. A. Fry for sending us the GFP-POC1A construct and Dr. I Hoffmann and Dr. E. Nigg for antibodies. This work is supported by grants of the Deutsche Forschungsgemeinschaft (DFG) to E.S. (DFG Schi295/8-2).

## Author contributions

C.S. performed most experiments. E.S.A. performed preliminary experiments. C.S. and E.S.A. generated constructs and KO cell lines. M.W. performed cloning, protein expression and protein purification from insect cells. A.N. performed EM, negative staining and data acquisition. T.H. and S.E. provided the AlphaFold2 predictions and interaction maps and helped in discussing and analysing them. P.P. contributed to IP experiments. C.S. designed all the experiments with input of E.S., C.S. and E.S. wrote the manuscript.

## Funding

## Competing interests

The authors declare no competing interests.
