## [Transparent Peer Review file · Nature Communications]

An interaction network of inner centriole proteins organised by POC1A-POC1B heterodimer crosslinks ensures centriolar integrity

Corresponding Author: Professor Elmar Schiebel

Version 0:

Reviewer comments:

Reviewer #1

(Remarks to the Author)

Sala et al REVIEW (Nat Comm)

This study by Sala and colleagues aims to understand the role of Poc1 and other inner scaffold molecules of the centriole lumen in stabilizing human centrioles. Extending previous work, the authors discover a Poc1A-Poc1B heterodimer to participate in organizing the inner scaffold of the centriole lumen. This study identifies an important network of molecules and their relative interactions for the inner lumen and, as previously shown, exemplifies the important stability of the centriole. They also show this to be true for centriole stability during mitosis. The experiments are nicely performed, and this is an important contribution to the field that both establishes functions for the two human Poc1 paralogs and their coordinated roles with Poc5 and Centrin. Below, I outline several points that the authors may consider before publication, most of this is a request for further discussion of how the data fit with the field.

MAJOR

Unfortunately, I think the model in Fig7F is backwards? In this image, Poc1A is near the centriole MT wall and Poc1B in the lumen. This caused severe confusion. Ultimately, I believe that this is backwards, and I have reviewed the manuscript as such.

I believe there should be a greater discussion of redundant pathways for the reported results. Because the authors observed only partial loss of proteins like Poc5, MDM1, CCDC15, FAM161 upon loss of the Poc1 paralogs.

Many of the interactions predicted from AlphaFold2 appear to be qualitative. Can the authors place confidence numbers to these interactions? In particular, can this be performed for specific domain interactions to justify the biological experiments. Why might some organisms only have a single Poc1 gene and vertebrates have two? How do the data of human Poc1 reported in this study influence our understanding of Poc1 studies in Chlamydomonas, Paramecium, and Tetrahymena?

MINOR

Line46-7 – Chlamy isn't a ciliate. Consider using "protist".

Line57 – Poc1 was found around the same time by the Winey lab (referenced) and the Marshal lab (no ref). Please include the Marshal ref (PMID: 15964273).

Line68 – PMID 36287828 should be included with refs.

Discussion of how and when Poc1 paralogs might incorporate into the centriole architecture to promote this organization would improve the manuscript. For example, does Poc1A incorporate into the inner junction earlier than Poc1B? How does this incorporation occur relative to Centrin and Poc5?

It would be interesting if the authors were to include experiments to test potential roles of WDR90 at the inner junction as was described in the Introduction. At the least, perhaps the authors can discuss this point.

Line125- protein > proteins

Fig1K – separation of colors are hard to appreciate in the figure. Please use colors with greater contrast.

Fig1I – indicate that this is colocalization with α -tubulin. This is also true for other figures in the manuscript (Fig2F, Fig S5I, S3.G, K, M, O,

Line252- The Poc1Ab and POC1Ba chimeras were not clearly explained. Perhaps referencing model figure in Fig 1L would improve this. Similarly, model of the Intra domain would also make this section clearer.

FigS5I-control figures need to be included.

Fig3H and 3I – clearer labeling of the figures are required to understand what this is.

Line214- Please clarify the statement “the scheme of POC5-Centrin and FAM161A”. I’m not sure what this means?

Fig 5E needs controls.

Line301: “predicated” should be “predicted”

Line395: The de novo assembly in poc1KO centriole loss is interesting and also consistent with a similar new assembly found in Tetrahymena when poc1 is absent. PMID: 26700722

SupFig8G: I believe the data supporting the idea of centriole overduplication was revealed only in this figure graph. Here, it was hard to appreciate that there is indeed centriole amplification. Can the authors show images of this?

Line430: I think “shows” should be suggests.

Line442: I think it would be clearer and more convincing to show the control in this image rather than referring the reader back to Fig2.

Line444: This image should be with alpha-tubulin (as in prior figures) and not gamma-tubulin.

Fig6G: heading says “mitotic spindle configuration in different KO cell lines”. Which KOs? Are all these images from poc1KO? Is there a control? Please indicate the relative frequencies of each event?

Line470-472: Poc1 interaction with inner scaffold is established in PMID: 38743010

SuppFig9E: complex is misspelled.

The figure text throughout the manuscript is small and hard to read.

Fig4H: Should CT be C-termA?

Fig5C was unclear. Can the authors please explain this figure more clearly.

Fig 6A: The loss in specific triplets was also found in Tetrahymena poc1 mutants (PMID: 27251062)

Fig7A/B: For comparison, it would be great to have the figures at the same scale.

Reviewer #2

(Remarks to the Author)

Centrioles are the core unit of the centrosome, the major microtubule organizing center. Although considerable advances have been made in understanding centriole composition and structure, how proteins of the centriole inner scaffold complex and the microtubule wall collaborate to preserve centriole integrity remains to be determined.

In this study, Sala et al. address important questions relating to centriole integrity. Specifically, they examine the interactions of POC1A and POC1B with each other, as well as with the inner scaffold protein POC5 and the microtubule-binding proteins FAM161A and MDM1. The authors propose that these interactions are mediated by the WD40 and/or C-terminal coiled-coil domains of the POC1A/B proteins. They propose that POC1A and B heterodimerize through their coiled-coil domains and localize to the centriole lumen. The WD40 domain of POC1B targets the heterodimer to the microtubule wall, while the WD40 domain of POC1A interacts with the POC5-Centrin inner scaffold protein complex. This offers new insight into how POC1 proteins contribute to organizing the complex interactions between the centriole microtubule wall and the inner scaffold protein complex.

Overall, this manuscript provides a valuable contribution to the centriole field. However, some alterations (detailed below) are needed to improve the presentation of the results and the coherence of the conclusions.

General comments:

- It would be helpful to review the paper for consistency in terminology, such as species names and how the references are presented. Moreover, the terminology surrounding the ‘mother/daughter/parent’ centriole should be better defined. Daughter centriole is sometimes used in the field to refer to the pro-centriole and sometimes to refer to the young mother centriole.
- The organization of the figures could be improved. I recommend restructuring the figure layout to follow a more logical sequence that aligns with the text’s narrative. Additionally, increasing the font size in the figures would help the reader. Some of the figure references in the text need to be clarified.
- A few figures show representative images for the edited/manipulated cell lines only without showing a direct comparison with the control cell line. Moreover, figure 7E only shows the quantification but not the representative U-ExM images from where those quantifications were derived.
- The authors claim they analyze centrioles from cells in specific cell cycle phases. How were these phases determined? Was pro-centriole length used to determine the cell cycle phase?
- The methods section needs to include how data analysis was performed. For example, it is unclear how the centriole coverage analysis by U-ExM or the centriole intensity analysis by IF was performed.

Specific comments by figure:

Figure1:

1G/H: The quantification of POC1B has a large spread. Are there differences in diameter at the proximal or distal regions of the centriole?

1J: Images and quantifications could be ordered from the distance to the centriole MT wall. Also, the color scheme should be the same as 1K to enable the reader to make direct comparisons.

Figure2 and respective supplementary figure:

2F: Does FAM161A lose a proximal centriole signal in POC1B-/- cell lines?

“POC1A plays a crucial role in the binding efficiency and compartmentation of inner centriolar proteins.” – what do the authors mean by the binding efficiency, and where is this represented?

Figure 3:

3G/H/I: this experiment seems to have only one replicate. Due to the importance of this experiment for the conclusion of this section, it would be important to repeat this experiment and show that the POC1A WD40 binds to POC5 more strongly than the POC1B WD40.

Figure 6:

S8A-C: keep the color scheme the same between graphs for easier visualization.

6G: Representative images of the different edited/manipulated cell should be shown.

Figure 7:

7E: representative images need to be shown. U-ExM of centrioles from the POXA/B KO cells would be very useful to evaluate the conclusions made concerning the POC1 double depletion and centriole biogenesis.

The colors for POC1A and POC1B appear to be swapped.

Supplementary

S3G: The authors claim that CEP44 exhibits a distal extension in POC1B^{-/-} centrioles. I don't see evidence for this in the representative images or quantifications.

S4I: positive control for interaction should be shown to show the detection method is working

S5: It would be useful to investigate whether MDM1 and CCDC15 are altered in POC5^{-/-} cells.

Points that could use a more detailed discussion:

Discuss the differences in the centriole MT wall structure across different species and how this relates to the presence of one or two POC1 isoforms.

POC1A loss leads to elongation of CEP135, CEP295, and CEP44 staining. Can the authors discuss this observation?

How does this fit in their model concerning POC1A interactions/localization more towards the lumen of the centriole?

Reviewer #3

(Remarks to the Author)

Version 1:

Reviewer comments:

Reviewer #1

(Remarks to the Author)

The authors have addressed my concerns.

Reviewer #2

(Remarks to the Author)

The authors have addressed all the comments and suggestions which significantly enhanced the clarity of the paper. They have strengthened their analyses and provided additional data where necessary. The paper now presents a more comprehensive and cohesive argument, which we believe will contribute valuable insights to the field of centriole biology. Thus, we recommend this study for publication.

Reviewer #3

(Remarks to the Author)

27. Sep. 2024

Rebuttal letter

Nature Communications manuscript NCOMMS-24-40532-T

We would like to thank all three reviewers for their helpful and positive comments. As you can see below, we have addressed all the comments that were raised.

REVIEWER COMMENTS

Reviewer #1 (Remarks to the Author):

Sala et al REVIEW (Nat Comm)

This study by Sala and colleagues aims to understand the role of Poc1 and other inner scaffold molecules of the centriole lumen in stabilizing human centrioles. Extending previous work, the authors discover a Poc1A-Poc1B heterodimer to participate in organizing the inner scaffold of the centriole lumen. This study identifies an important network of molecules and their relative interactions for the inner lumen and, as previously shown, exemplifies the important stability of the centriole. They also show this to be true for centriole stability during mitosis. The experiments are nicely performed, and this is an important contribution to the field that both establishes functions for the two human Poc1 paralogs and their coordinated roles with Poc5 and Centrin. Below, I outline several points that the authors may consider before publication, most of this is a request for further discussion of how the data fit with the field.

We thank reviewer #1 for the positive comments and the remark to discuss in more detail the redundancy of the POC1 proteins and what we can conclude from the human study for protists that have only one POC1 protein. We modified the manuscript to address these questions and to make our statements clearer for the audience. Particularly, we have extended the discussion describing in detail inner scaffold proteins in Tetrahymena as an example (see Discussion, lines 588-616).

MAJOR

Unfortunately, I think the model in Fig7F is backwards? In this image, Poc1A is near the centriole MT wall and Poc1B in the lumen. This caused severe confusion. Ultimately, I believe that this is backwards, and I have reviewed the manuscript as such.

This was indeed a mistake in the model. We apologize for this and the caused confusion and would like to thank the reviewer for considering it backwards during the reviewing process. We changed it now.

I believe there should be a greater discussion of redundant pathways for the reported results. Because the authors observed only partial loss of proteins like Poc5, MDM1, CCDC15, FAM161 upon loss of the Poc1 paralogs.

We now discuss redundancy between POC1A and POC1B, FAM161 and MDM1, POC1 and WDR90/POC16. Lines 556-565: "An additional strategy for ensuring proper centriole duplication is the functional redundancy of structural proteins. This helps safeguard the process by preventing mutations in a single gene from completely disrupting centriole duplication. Although POC1A and POC1B have specialized roles, this study demonstrates that they are also partially redundant in their function, as the simultaneous loss of both genes results in more severe defects than the loss of either

gene alone. Additionally, MDM1 and FAM161A may function redundantly in connecting the inner scaffold to the centriole MT wall. Similarly, POC1 and WDR16/WDR90 could have overlapping roles in maintaining A-B centriole MT junctions, though further investigation is needed to test this possibility (see below).”.

Many of the interactions predicted from AlphaFold2 appear to be qualitative. Can the authors place confidence numbers to these interactions? In particular, can this be performed for specific domain interactions to justify the biological experiments.

As suggested, we added PAE plots and the scores of the ensemble interaction maps in Supplementary Figs. 6, 7, 9, 10, 11, 12 and 14. In addition, in the new Supplementary Fig. 5 we show the analysis pipeline how these scores were calculated for POC1A-POC1B interaction as an example. With this approach potential interactions between specific domains can be revealed.

Why might some organisms only have a single Poc1 gene and vertebrates have two? How do the data of human Poc1 reported in this study influence our understanding of Poc1 studies in Chlamydomonas, Paramecium, and Tetrahymena?

We added these points to our Discussion.

Lanes 551-556: “The duplication of the POC1 gene likely occurred early in metazoan evolution, as most vertebrate species possess two POC1 paralogues, POC1A and POC1B. As we have shown in this study, the specialization of POC1A and POC1B enhances specificity, potentially leading to more efficient assembly and maintenance of centrioles. This is particularly important for organisms that rely on numerous cell divisions for development, each of which depends on efficient spindle formation and functional centrosomes/centrioles.”

Lanes 600-605: “Human POC1A and POC1B dimerize through their C-terminal coiled-coil domains and this ability may extend to POC1 proteins in other organisms as indicated by the conservation of the C-terminal coiled-coil. POC1-POC1 interaction would allow one WD40 domain to bind to centriole MTs while the other recruits inner scaffold proteins. Consistently, in Tetrahymena and humans, proper localisation of the inner scaffold protein FAM161A relies on the presence of POC1¹¹.”

MINOR

Line46-7 – Chlamy isn't a ciliate. Consider using “protist”.

Thank you for pointing this out. We changed it in the manuscript to “protist” as suggested by reviewer #1.

Line57 – Poc1 was found around the same time by the Winey lab (referenced) and the Marshal lab (no ref). Please include the Marshal ref (PMID: 15964273).

We included now this important reference as suggested.

Line68 – PMID 36287828 should be included with refs.

Thank you very much for pointing this out. We included this reference as suggested.

Discussion of how and when Poc1 paralogs might incorporate into the centriole architecture to promote this organization would improve the manuscript. For example, does Poc1A incorporate into the inner junction earlier than Poc1B? How does this incorporation occur relative to Centrin and Poc5? *As suggested, we discuss the timing of recruitment of inner scaffold components on lines 525-536: “The timing of inner scaffold protein recruitment to newly assembled procentrioles varies. Both POC1 proteins are recruited to procentrioles at a length of approximately 120-140 nm, though it remains possible that one protein is recruited before the other. POC1B interacts with CEP44, which is found early at the procentriole¹⁷, and extends further proximal than POC1A, making it likely that POC1B's*

recruitment to the centriole is slightly ahead of POC1A's. Unlike POC1, both POC5 and FAM161A have been shown to localise to procentrioles once they reach a length of 160 nm. This may also apply to the central pool of Centrin, which is dependent on POC5 for its recruitment^{18,32}. These findings suggest that POC1A/B are recruited earlier than POC5 and FAM161A during procentriole assembly, consistent with the POC1A-POC1B heterodimer being essential for POC5 targeting to centrioles. It would be interesting to explore the timing of CCDC15 recruitment relative to other inner scaffold proteins."

It would be interesting if the authors were to include experiments to test potential roles of WDR90 at the inner junction as was described in the Introduction. At the least, perhaps the authors can discuss this point.

Thank you very much for pointing this out. We have performed centriole localisation studies based on the published WDR90 antibodies. This experiment is now shown in Supplementary Fig. 3a and b and described in Results. It shows that WDR90 centriole localisation is reduced in POC1A^{-/-} but not in POC1B^{-/-} cells. A stronger impact on WDR90 localisation at centrioles was observed in POC5^{-/-} cells. We also tried expansion microscopy analysis with WDR90 antibodies. However, we are not satisfied with the quality of the centriole staining by the published WDR90 antibody under U-ExM condition and decided not to show these images in the manuscript. The WDR90 signal was dotted (1-2 dots per centriole in wild type and mutants) and it was very difficult to judge whether the localisation pattern was changed in POC1A^{-/-} cells.

Line125- protein > proteins

We changed this typo now in the manuscript.

Fig1K – separation of colors are hard to appreciate in the figure. Please use colors with greater contrast.

We changed colors in Fig. 1k and made the spacing bigger as suggested.

Fig1I – indicate that this is colocalization with α -tubulin. This is also true for other figures in the manuscript (Fig2F, Fig S5I, S3.G, K, M, O,

We now indicate this for all relevant U-ExM images as α -Tub/M (for merged) and hope that the figures are now much easier to understand.

Line252- The Poc1Ab and POC1Ba chimeras were not clearly explained. Perhaps referencing model figure in Fig 1L would improve this. Similarly, model of the Intra domain would also make this section clearer.

We added an explanation and a reference back to Fig. 1l as suggested by the reviewer and hope that the concept of the POC1 chimeras and the Intra region of POC1B is now more understandable.

FigS5I-control figures need to be included.

Thank you for pointing this out. We now included images from the control cell line.

Fig3H and 3I – clearer labeling of the figures are required to understand what this is.

For Fig. 3h and I we added now Headers to make it clearer what the quantification shows and that it derives from one representative experiment (added to figure legend). We included also in the figure legend that this experiment was conducted three times independently (total number of experiments: N=3) but one representative result is shown due to variations in expression across the different experiments and to avoid multiple normalisation steps. Each prey/bait ratio of the experiments is listed in the source data for Fig. 3f, h and i. and the corresponding blots are shown in the supplementary source data- immunoblots. This we also indicated in the legend of Fig. 3h and i.

Line214- Please clarify the statement “the scheme of POC5-Centrin and FAM161A”. I’m not sure what this means?

We revised the phrasing on lines 205-207 to “Furthermore, the centriole-inside localisation of γ -tubulin and HAUS4²⁴, was disrupted in POC1A^{-/-} cells but remained unaffected in POC1B^{-/-} cells (Fig. 2f, i; Supplemental Fig. 3i, j).”

Fig 5E needs controls.

We show the FRET efficiency only for the donor:acceptor sample, as we only observe a decrease in lifetime of the donor in the presence of the acceptor.

Line301: “predicated” should be “predicted”

We changed this typo in the revised manuscript.

Line395: The de novo assembly in poc1KO centriole loss is interesting and also consistent with a similar new assembly found in Tetrahymena when poc1 is absent. PMID: 26700722

Lines 578-581 of the Discussion: “However, Tetrahymena POC1 plays a critical role in stabilising basal bodies when they are subjected to mechanical stress from cilia beating probably because it reinforces the triplet MTs by localizing to the A-B and B-C MT inner junctions¹¹. It also plays a role in regulating basal body number.”

SupFig8G: I believe the data supporting the idea of centriole overduplication was revealed only in this figure graph. Here, it was hard to appreciate that there is indeed centriole amplification. Can the authors show images of this?

Reviewer #1 is probably referring to former Suppl. Fig. 8f, because there we show IF images of centriole amplification. We replaced it with a better image for the POC1A^{-/-} cell line (now Supplementary Fig. 13f).

Line430: I think “shows” should be suggests.

We changed it in the manuscript as suggested by reviewer #1.

Line442: I think it would be clearer and more convincing to show the control in this image rather than referring the reader back to Fig2.

Images from the control cell line are now included in the Fig. 7d.

Line444: This image should be with alpha-tubulin (as in prior figures) and not gamma-tubulin.

We apologize that our labelling was misleading. In Fig. 7d a triple staining was performed where we stained the HA-tagged POC5, γ -tubulin (to assess the rescue ability of the POC5 construct) and α -tubulin that acts throughout the paper as a centriolar wall marker. As in a previous comment from reviewer #1, we indicated now in the merged channel “M” that the grey signal corresponds to α -tubulin.

Fig6G: heading says “mitotic spindle configuration in different KO cell lines”. Which KOs? Are all these images from poc1KO? Is there a control? Please indicate the relative frequencies of each event?

We apologize that we did not elaborate this clearer. Fig. 6g was a representative summary of the mitotic spindle phenotypes that were observed in the POC1^{-/-} as well as the control cell line, to establish our definition of each configuration. We have now changed this by showing the most prominent phenotypes for Control/wild type, POC1A^{-/-}, POC1B^{-/-} and POC1A/B^{-/-} cells in Fig. 6g. In Fig. 6h, we show the quantification of these phenotypes.

Line470-472: Poc1 interaction with inner scaffold is established in PMID: 38743010

We included this reference in the manuscript.

SuppFig9E: complex is misspelled.

We corrected this typo in the Supplementary Figure.

The figure text throughout the manuscript is small and hard to read.

We made changes throughout the manuscript to make the manuscript more reader friendly.

Fig4H: Should CT be C-termA?

We changed it to "C-termA" for consistency.

Fig5C was unclear. Can the authors please explain this figure more clearly.

We added an explanation for the phasor plot of the FLIM-FRET experiment in the main text (lines 356-359) and added a description in the corresponding Fig. 5 legend.

Fig 6A: The loss in specific triplets was also found in Tetrahymena poc1 mutants (PMID: 27251062)

This is indeed an important paper that also goes nicely hand in hand with our findings. The reference is now included.

Fig7A/B: For comparison, it would be great to have the figures at the same scale.

For better comparison, we added an image with the same scale as suggested by reviewer #1 in Fig.7b. We also compare both complexes in Fig. 7c.

Reviewer #2 (Remarks to the Author):

Centrioles are the core unit of the centrosome, the major microtubule organizing center. Although considerable advances have been made in understanding centriole composition and structure, how proteins of the centriole inner scaffold complex and the microtubule wall collaborate to preserve centriole integrity remains to be determined.

In this study, Sala et al. address important questions relating to centriole integrity. Specifically, they examine the interactions of POC1A and POC1B with each other, as well as with the inner scaffold protein POC5 and the microtubule-binding proteins FAM161A and MDM1. The authors propose that these interactions are mediated by the WD40 and/or C-terminal coiled-coil domains of the POC1A/B proteins. They propose that POC1A and B heterodimerize through their coiled-coil domains and localize to the centriole lumen. The WD40 domain of POC1B targets the heterodimer to the microtubule wall, while the WD40 domain of POC1A interacts with the POC5-Centrin inner scaffold protein complex. This offers new insight into how POC1 proteins contribute to organizing the complex interactions between the centriole microtubule wall and the inner scaffold protein complex.

Overall, this manuscript provides a valuable contribution to the centriole field. However, some alterations (detailed below) are needed to improve the presentation of the results and the coherence of the conclusions.

General comments:

- It would be helpful to review the paper for consistency in terminology, such as species names and how the references are presented. Moreover, the terminology surrounding the 'mother/daughter/parent' centriole should be better defined. Daughter centriole is sometimes used in the field to refer to the procentriole and sometimes to refer to the young mother centriole.

In order to avoid confusion, we have now defined the terms daughter centriole and procentriole. First, in the Introduction from lines 53-57 we introduce centriole duplication: “Centrioles duplicate once per cell cycle starting in G1/S phase. A new centriole, known as the procentriole, begins to assemble at the proximal end of each of the two existing mother centrioles. The procentriole elongates during G2 and matures by the end of mitosis into a centrosome (centriole with pericentriolar proteins), at which point it disengages from the mother centriole².”

Second, throughout the text, we use the terms mother centriole or procentrioles. We do not use the term “daughter centriole”.

- The organization of the figures could be improved. I recommend restructuring the figure layout to follow a more logical sequence that aligns with the text's narrative. Additionally, increasing the font size in the figures would help the reader. Some of the figure references in the text need to be clarified.

To address this point, we have restructured the part explaining the IF data in Fig. 2. We first focus on the inner scaffold proteins and next describe proximal end proteins (CEP44, CEP135, CEP192) and Sas-6. In addition, we show in Supplemental Fig. 5 an example of our analysis pipeline. Furthermore, we have rearranged Supplementary Figs. and added the PAE plots.

- A few figures show representative images for the edited/manipulated cell lines only without showing a direct comparison with the control cell line. Moreover, figure 7E only shows the quantification but not the representative U-ExM images from where those quantifications were derived.

As requested, we included in all figures the control cell line to show direct comparison and to avoid that the reader has to go back to a previous figure. Specifically, to Fig. 7d we have added the U-ExM image of the “wild type” control (POC5^{-/-}/POC5-HA).

- The authors claim they analyze centrioles from cells in specific cell cycle phases. How were these phases determined? Was procentriole length used to determine the cell cycle phase?

Indeed, it is difficult to be sure about the cell cycle phase of the cell based on expansion microscopy images. To avoid this problem, we have measured the length of the procentriole when POC1A and POC1B are recruited. This data is now given in the Result section, lines 143-146: “U-ExM revealed that in RPE1 cells, the recruitment of POC1A and POC1B to newly formed procentrioles occurs when procentrioles are still relatively short, measuring between approximately 120 and 140 nm (Fig. 1d).”.

- The methods section needs to include how data analysis was performed. For example, it is unclear how the centriole coverage analysis by U-ExM or the centriole intensity analysis by IF was performed.

We included measurement and analysis procedure of U-ExM in the method section under “Length and diameter measurements of longitudinal centrioles and top view centrioles”. For the signal intensity measurements of normal IF data we included this part under “Immunofluorescence and signal intensity measurement” referring to a semi-automated Macro that was published from our group by Hata et al. in Nat Cell Bio from 2019.

Specific comments by figure:

Figure1:

♣ 1G/H: The quantification of POC1B has a large spread. Are there differences in diameter at the proximal or distal regions of the centriole?

The distribution of the signal diameter and the distance from the inner scaffold protein to the centriole microtubule signal is relatively broad (Fig. 1h-j). Notably, for POC1B, this distribution is even wider compared to the other measurements. Despite this variation, the differences observed between POC1A, POC1B, and other inner scaffold proteins are statistically significant.

The broader spread can likely be attributed to a slight decrease in the diameter between the proximal and central regions of the centrioles (Steib et al., 2020, eLife). Additionally, POC1B extends the POC1A-associated centriole region, as illustrated in Fig. 1f. This suggests that POC1B exhibits a distinct architecture in the most proximal and distal regions of the centriole compared to the region where it overlaps with POC1A. In the U-ExM measurements, it is not possible to determine the exact location within the centriole being measured (for technical reasons, we can only measure in two channels and one of them is α -tubulin). Thus, some POC1B images may correspond to regions lacking POC1A, which explains the differing behavior of POC1B in these images.

In response to this comment, we added in the Discussion, lanes 511-513: "It is important to note that POC1B in the proximal and distal ends of the centrioles, where it does not overlap with POC1A, may exhibit a different organizational structure compared to the overlapping region of POC1A in the central part of the centriole."

♣ 1J: Images and quantifications could be ordered from the distance to the centriole MT wall. Also, the color scheme should be the same as 1K to enable the reader to make direct comparisons. *We changed the order of the proteins based on the distance to the MT wall in the U-ExM images. However, we retained the original data points for POC1A and POC1B, changing only the order of the inner scaffold proteins based on their increasing distance from the MT wall to facilitate easier comparison and highlight significant differences. CEP44 has a specific role because it's a proximal, microtubule binding protein and we wanted to point out that MT-binding proteins are closer to the MT wall.*

Figure 2 and respective supplementary figure:

♣ 2F: Does FAM161A lose a proximal centriole signal in POC1B^{-/-} cell lines?

The loss of the proximal FAM161A signal appears more frequently in POC1B^{-/-} cells compared to control and POC1A^{-/-} cells. However, we did not investigate this further, as our focus was on the inner scaffold region.

♣ "POC1A plays a crucial role in the binding efficiency and compartmentation of inner centriolar proteins." – what do the authors mean by the binding efficiency, and where is this represented? *Our original sentence might be misleading. What we want to express is that POC1A functions in effectively recruiting the respective proteins and also ensures their proper localisation within the centriole, as we can see in the POC1A KO (POC5 and FAM161A signal is not completely lost but they lose their defined localisation). We changed the sentence to avoid any misunderstandings. Lines 230-232: "In summary, POC1A plays a crucial role in the recruitment and proper compartmentation of inner centriolar proteins POC5, FAM161A, MDM1, CCDC15, and the inner centriolar pool of γ -tubulin and HAUS4."*

Figure 3:

♣ 3G/H/I: this experiment seems to have only one replicate. Due to the importance of this experiment for the conclusion of this section, it would be important to repeat this experiment and show that the POC1A WD40 binds to POC5 more strongly than the POC1B WD40.

We apologize for not pointing this out clearer and for the misleading impression. The blot in Fig. 3g is a representative blot out of 3 independent experiments. The quantifications in Fig. 3h and 3i correspond to the representative blot in Fig. 3g. Pooling of the ratios led to high error bars, coming from variations in expression between the different experiments and we wanted to avoid multiple normalisation steps, but within the experiments it's consistent that the WD40A shows a stronger binding to POC5, than WD40B. The same is the case for the chimeric versions shown in Fig. 3i. We indicated this now in the legend of Fig. 3g: „Quantifications show the result from one representative

experiment. Total number of independent experiments: N=3. Although the outcomes of the experiments were identical, there was significant variation between them. Therefore, we present only one quantified result.”.

Although, we do not show the ratios of all experiments in the Main Figure, we listed for each experiment the calculated prey/ratios in the respective source data of Fig. 3h and i. The blots where the quantification derives from is shown in the supplementary source data-immunoblots, this we also indicated in the legend of Fig. 3h and i.

Figure 6:

♣ S8A-C: keep the color scheme the same between graphs for easier visualization.

We changed the colors for easier comparison as suggested by reviewer #2.

♣ 6G: Representative images of the different edited/manipulated cell should be shown.

We apologize that we did not elaborate this clearer. Fig. 6g was a representative summary of the mitotic spindle phenotypes that were observed in the POC1^{-/-} as well as the control cell line, to establish our definition of each configuration. We have now changed this by showing the most prominent phenotypes for Control /wild type, POC1A^{-/-}, POC1B^{-/-} and POC1A/B^{-/-} cells in Fig. 6g. In Fig. 6h, we show the quantification of these phenotypes.

Figure

7:

♣ 7E: representative images need to be shown. U-ExM of centrioles from the POXA/B KO cells would be very useful to evaluate the conclusions made concerning the POC1 double depletion and centriole biogenesis.

Thank you very much for pointing this out. We included representative U-ExM images for the top view centrioles in Fig. 7f. The corresponding quantification is shown in Fig.7g.

U-ExM of centrioles from POC1A/B double knockout cells (POC1A/B^{-/-}) are impossible to obtain as can be seen in the EM images that shows only remnants of centrioles in Suppl. Fig. 13j.

♣ The colors for POC1A and POC1B appear to be swapped.

The colors were indeed swapped. We apologize for this and the confusion that caused this. We now changed the colors.

Supplementary

S3G: The authors claim that CEP44 exhibits a distal extension in POC1B^{-/-} centrioles. I don't see evidence for this in the representative images or quantifications.

We replaced the image for the CEP44 staining in POC1B^{-/-} cells with an image that reflects the average behavior better (see Supplementary Fig. 4g). This extension is also clearly seen in the quantification (Supplementary Fig. 4j).

S4I: positive control for interaction should be shown to show the detection method is working

We repeated the FLAG-IP experiment including the POC1A-FLAG as a positive control. These modified data are now shown in Suppl. Fig. 7e. Although POC5-HA is relatively weakly expressed in this experiment, it is efficiently enriched by the POC1A-FLAG IP. C-TermA/B-FLAG do not interact with POC5-HA in this experimental set up.

S5: It would be useful to investigate whether MDM1 and CCDC15 are altered in POC5^{-/-} cells.

As suggested, we added this experiment to Suppl. Fig. 8i.

Points that could use a more detailed discussion:

♣ Discuss the differences in the centriole MT wall structure across different species and how this relates to the presence of one or two POC1 isoforms.

These differences between MT wall structure and POC1 was added to the Discussion.

Lanes 566-570: "Centriole inner scaffold proteins are conserved across a wide range of species, from protists to mammals, which raises the question of whether our model is broadly applicable. While the overall architecture of centrioles is largely conserved among different species, there are significant variations, particularly concerning the cartwheel structure and the binding of inner centriole proteins to the microtubule wall³⁵."

From Lanes 551-556 we discuss why metazoan might have two POC1 genes: "The duplication of the POC1 gene likely occurred early in metazoan evolution, as most vertebrate species possess two POC1 paralogues, POC1A and POC1B. As we have shown in this study, the specialization of POC1A and POC1B enhances specificity, potentially leading to more efficient assembly and maintenance of centrioles. This is particularly important for organisms that rely on numerous cell divisions, each of which depends on efficient spindle formation and functional centrosomes/centrioles."

♣ POC1A loss leads to elongation of CEP135, CEP295, and CEP44 staining. Can the authors discuss this observation? How does this fit in their model concerning POC1A interactions/localization more towards the lumen of the centriole?

We extended the Discussion as suggested by the reviewer. Lanes 537-550:

"Inside the centriole, various substructures, including the cartwheel and the A-C linker, coexist alongside the inner scaffold, each exhibiting distinct lengths. These substructures may act as barriers that restrict the extension of neighboring structures. In POC1A^{-/-} cells, which have decreased levels of the MT-binding proteins FAM161 and MDM1 within the centrioles, proximal proteins extend into the central and distal regions of the centriole. The reduction of FAM161 and MDM1 exposes free MT-binding sites, allowing them to be occupied by other MT-binding proteins such as CEP44, CEP135, and CEP295, which consequently leads to their extension. A similar phenotype was observed in RPE1 TUBD1^{-/-} and TUBE1^{-/-} cells³³. Centrioles lacking δ - and ϵ -tubulin contain only A-MTs, fail to recruit POC5, and exhibit an elongated localisation of proximal proteins^{33,34}. However, unlike TUBD1^{-/-} and TUBE1^{-/-} cells³³, POC1A^{-/-} cells do not show an extension of the cartwheel protein SAS-6, suggesting a different mechanism for restricting the cartwheel's length. This mechanism might depend on the presence of MT triplets, which can still form in POC1A^{-/-} cells but are absent in the singlet-MT-containing TUBD1^{-/-} and TUBE1^{-/-} cells³⁴."

Reviewer #3 (Remarks to the Author):

We would like to thank reviewer #3 for positively reviewing our manuscript.